# The SPOC domain is a phosphoserine binding module that bridges transcription machinery with co- and post-transcriptional regulators

Lisa-Marie Appel [1,2,3], Vedran Franke [4], Johannes Benedum [1,2,3,5], Irina Grishkovskaya [6], Xué Strobl [3,5], Anton Polyansky [6], Gregor Ammann [7], Sebastian Platzer [3], Andrea Neudolt [3], Anna Wunder[3], Lena Walch[3], Stefanie Kaiser[7], Bojan Zagrovic [6], Kristina Djinovic-Carugo [6,8,9], Altuna Akalin [4] & Dea Slade [1,2,3] ✉

The heptad repeats of the C-terminal domain (CTD) of RNA polymerase II (Pol II) are extensively modified throughout the transcription cycle. The CTD coordinates RNA synthesis and processing by recruiting transcription regulators as well as RNA capping, splicing and 3′end processing factors. The SPOC domain of PHF3 was recently identified as a CTD reader domain specifically binding to phosphorylated serine-2 residues in adjacent CTD repeats. Here, we establish the SPOC domains of the human proteins DIDO, SHARP (also known as SPEN) and RBM15 as phosphoserine binding modules that can act as CTD readers but also recognize other phosphorylated binding partners. We report the crystal structure of SHARP SPOC in complex with CTD and identify the molecular determinants for its specific binding to phosphorylated serine-5. PHF3 and DIDO SPOC domains preferentially interact with the Pol II elongation complex, while RBM15 and SHARP SPOC domains engage with writers and readers of m$^6$A, the most abundant RNA modification. RBM15 positively regulates m$^6$A levels and mRNA stability in a SPOC-dependent manner, while SHARP SPOC is essential for its localization to inactive X-chromosomes. Our findings suggest that the SPOC domain is a major interface between the transcription machinery and regulators of transcription and co-transcriptional processes.

The SPOC (Spen orthologue and paralogue C-terminal) domain is a 15–20 kDa protein domain found across eukaryotic species from yeast to mammals[1]. SPOC domains form a distorted β-barrel structure comprising seven β-strands and a variable number of α-helices[2–5]. The six human SPOC-containing proteins can be divided into three groups based on their domain organization (Supplementary Fig. 1): Spen family, DIDO/PHF3, and SPOCD1. SHARP

(SMRT/HDAC1 associated repressor protein), RBM15, and RBM15B (RNA binding motif protein 15/15B) paralogues belong to the Spen family of proteins characterized by a series of N-terminal RRMs (RNA recognition motifs) and the C-terminal SPOC domain. DIDO (Death inducer obliterator) and PHF3 (PHD finger protein 3) share a different domain architecture comprising a PHD (Plant homeodomain), a TLD (TFIIS-like domain) and the SPOC domain. SPOCD1

(SPOC domain-containing protein 1) originated from a duplication of PHF3[6] but lacks the PHD.

SPOC-containing proteins are generally associated with transcription regulation, differentiation, and development[5,7–10]. The SPOC domain of SHARP is crucial for its repressive function in the Notch signaling pathway. It recruits corepressor complexes SMRT/NCoR-HDAC1 by binding to phosphorylated serine within the conserved LSD motif at the SMRT/NCoR C-terminus[3,11]. More recently, SHARP SPOC has been implicated in Xist lncRNA-mediated silencing during X-chromosome inactivation, but the mechanism remains elusive[12]. In comparison to SHARP SPOC, the SPOC domains of the other two Spen family members, RBM15 and RBM15B, have a weaker effect in repressing transcription when tethered to a promoter through Gal4-DBD[13]. Instead, RBM15 and RBM15B are involved in post-transcriptional regulation, mainly by influencing alternative splicing, m[6]A (N[6]-methyladenosine) RNA modification, and nuclear export[10,14–16]. Little is known about the function of their SPOC domains. RBM15 SPOC was shown to bind to the unstructured LPDSD motif of the histone H3K4me3 methyltransferase[17]. Additionally, RBM15 and RBM15B SPOC domains were shown to bind to the Epstein-Barr virus early protein EB2, which promotes the nuclear export of viral mRNAs[13].

We recently showed that PHF3 SPOC specifically binds the C-terminal domain (CTD) of the RNA polymerase II (Pol II) subunit RPB1 phosphorylated on serine-2 in tandem repeats[5]. The disordered CTD comprises up to 52 imperfect heptad repeats of the sequence YSPTSPS and is differentially modified throughout the transcription cycle[18,19]. Phosphorylation of serine-5 is a mark of the early stages of transcription, while productive elongation is linked with serine-2 phosphorylation[18]. Different phosphomarks are recognized by CTD reader domains, ensuring the timely recruitment of transcription regulators and RNA processing factors to the transcription machinery[19–21]. The SPOC domain is critical for PHF3-mediated regulation of transcription and mRNA stability of neuronal genes[5]. The PHF3 paralogue DIDO regulates splicing and is critical for stem cell self-renewal and differentiation[22–25], while SPOCD1 is required for the silencing of transposable elements through piRNA-mediated methylation[6]. However, the function of DIDO and SPOCD1 SPOC domains has remained unclear.

SHARP and PHF3 SPOC have been established as phosphoserine binding domains raising the question as to whether other SPOC domains also bind phosphorylated serine and how they contribute to protein interaction networks. Here we show that the SPOC domains of DIDO, SHARP, and RBM15 act as CTD reader domains that bind phosphorylated serines via conserved surface patches. We report the crystal structure of SHARP in complex with serine-5-phosphorylated CTD and determine the similarities and differences between SHARP-SMRT and SHARP-CTD interaction on the structural level. We further applied mass spectrometry to identify SPOC-dependent interactors of PHF3, DIDO, SHARP, and RBM15. Our findings establish Pol II elongation machinery as the focal point for PHF3 and DIDO SPOC interactions, while m[6]A writer and reader proteins are major targets for RBM15 and SHARP SPOC domains. Collectively, our results suggest that SPOC is a versatile phosphoserine binding module that spans the transcription machinery, and co- and post-transcriptional regulators.

## Results

### Conserved basic residues cluster to patches on the surface of SPOC

PHF3 SPOC has been established as a CTD reader domain[5], but it has remained unknown whether CTD binding is unique to PHF3 or if other SPOC domains possess the same ability. SPOC domains show an overall low level of sequence conservation, however, some key residues are conserved (Fig. 1a). These include an arginine residue (red asterisk in Fig. 1a) that is conserved in all human SPOC domains except SPOCD1 and is critical for the electrostatic anchoring of phosphorylated serine in CTD by PHF3 SPOC and in SMRT/NCoR by SHARP SPOC

(R1248 in PHF3, R3552 in SHARP, Figs. 1b, c and 3e)[3,5,11]. This arginine residue is part of a positively charged patch on the surface of SPOC. PHF3 SPOC has two such patches (Fig. 1b), while SHARP SPOC has one basic patch (Fig. 1c), which contains additional conserved lysine and arginine residues (Fig. 1a, conserved residues marked with colored squares). Although experimental structural information on DIDO SPOC is lacking, AlphaFold2 structure prediction[26,27] shows that the conserved residues cluster in patches on the domain surface similar to PHF3 SPOC (Fig. 1d), suggesting that these domains may have similar phosphoserine binding properties. We solved the crystal structure of RBM15 SPOC (to 1.45 Å resolution), which has a distorted β-barrel fold comparable to previously structurally characterized SPOC domains (Fig. 1e, Supplementary Fig. 2a–c, Supplementary Table 1). The conserved lysine and arginine residues cluster to a basic surface patch similar to that of SHARP SPOC (Fig. 1e, Supplementary Fig. 2b). Although RBM15 SPOC dimers were present in the asymmetric unit, SEC-MALS analysis revealed that RBM15 SPOC forms monomers in solution, which was also the case for other SPOC domains (Supplementary Fig. 2d–g). PHF3 SPOC showed a mixture of monomers and dimers (70% vs. 30%) (Supplementary Fig. 2d), while the previous analysis of full-length PHF3 revealed a purely monomeric form[5]. Phylogenetic analysis indicates that SPOCD1 originated from a duplication of PHF3[6]; however, only two out of four basic residues from PHF3 are also conserved in SPOCD1 (Fig. 1a), suggesting that SPOCD1 SPOC has lost the phosphoserine binding ability. Indeed, the structure predicted by AlphaFold2 reveals a much weaker positive charge in the surface patches of SPOCD1 SPOC (Fig. 1f). Furthermore, the distance between the patches is 36 Å, which is considerably further apart than the patches of PHF3 or DIDO SPOC (24 Å and 21 Å, respectively) and makes it highly unlikely that SPOCD1 SPOC could accommodate CTD phosphorylations in adjacent repeats.

### Basic surface patches mediate SPOC binding to phosphorylated serine

To test whether SPOC domains are universal CTD binders and determine their binding specificity, we expressed and purified the SPOC domains of the human proteins PHF3, DIDO, SHARP, RBM15, and SPOCD1 and performed fluorescence anisotropy assays to measure their binding to Atto488-labeled CTD-diheptapeptides (Fig. 2, Supplementary Figs. 3–7). To obtain a comprehensive picture of SPOC-CTD interactions, we measured binding affinities for a total of eleven peptides that were either unphosphorylated (CTD) or phosphorylated in one (1×S2P, 1×S5P, 1×S7P, 1×Y1P, 1×T4P) or both repeats (2×S2P, 2×S5P, 2×S7P, 2×Y1P, 2×T4P) (Supplementary Table 2). We first confirmed our previous finding that PHF3 SPOC preferentially binds to CTD phosphorylated at serine-2 in two adjacent repeats (2×S2P, $K_d = 0.42 \pm 0.02\,\mu M$, Fig. 2a). PHF3 did not bind to unphosphorylated or single phosphorylated CTD peptides and bound with lower affinity to other double phosphorylations (Fig. 2a, e and Supplementary Fig. 3; $K_d$ ranging from 29 to 130 μM). Similarly, the SPOC domain of the PHF3 paralogue DIDO showed the highest affinity for the 2×S2P peptide ($K_d = 4.8 \pm 0.6\,\mu M$), did not bind unphosphorylated or single phosphorylated CTD, and bound other double phosphorylated peptides with low affinity (Fig. 2b, e and Supplementary Fig. 4; $K_d$ ranging from 102 to 298 μM). Mutation of the conserved arginine residue to alanine greatly reduced the affinity to 2×S2P CTD ($K_d = 8.4 \pm 0.3\,\mu M$ for PHF3 SPOC R1248A, $22.4 \pm 2.1\,\mu M$ for DIDO SPOC R1096A; Fig. 2a, b), confirming the critical role of this residue in phosphoserine recognition. SPOCD1 SPOC did not bind to unphosphorylated, single or double phosphorylated CTD peptides (Supplementary Fig. 5). This indicates that SPOCD1 SPOC may indeed have lost the phosphoserine binding ability, although we cannot exclude that it binds phosphorylated interaction partners other than the CTD.

SHARP SPOC binds to a conserved LSD motif at the C-terminus of the transcriptional corepressors SMRT and NCoR in a serine

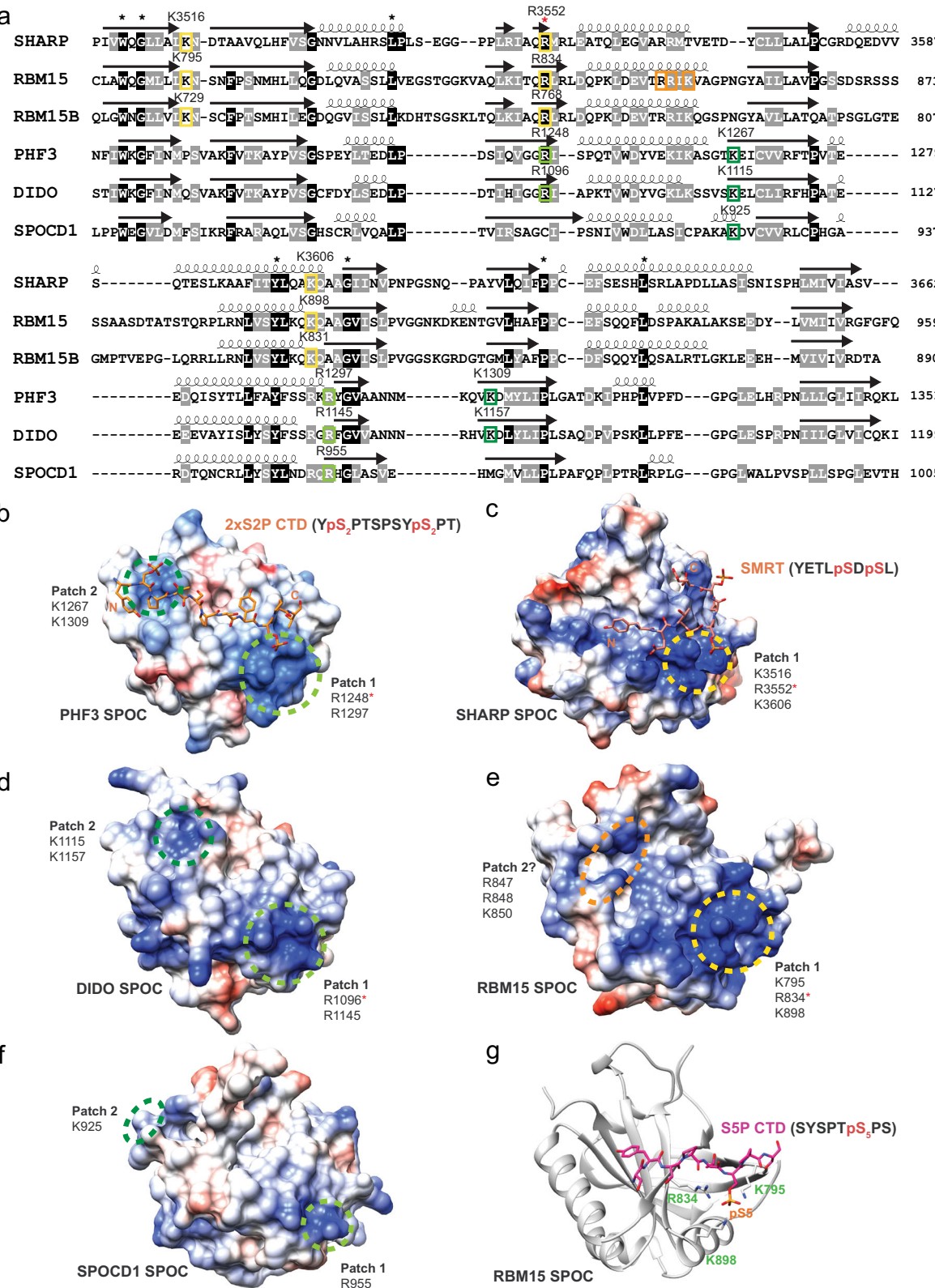

phosphorylation-dependent manner[2,3,11] (Figs. 1c and 3e). More recently, Pol II has been identified as a binding partner of SHARP SPOC[12], supporting the idea that SHARP SPOC might be a CTD binder akin to PHF3. Indeed, SHARP SPOC bound to CTD peptides phosphorylated at serine-5 in fluorescence anisotropy assays (Fig. 2c, e). It did not bind to unphosphorylated CTD and showed a very low affinity for other CTD phosphorylations (Fig. 2c, e and Supplementary Fig. 6). In contrast to PHF3 and DIDO, SHARP only required phosphorylation

in a single repeat; double phosphorylation in adjacent repeats barely affected the binding affinity ($K_d = 23.8 \pm 0.8\,\mu M$ for 1×S5P, $20.6 \pm 0.1\,\mu M$ for 2×S5P), which is in line with the fact that SHARP SPOC has only one conserved basic patch on its surface while PHF3 SPOC has two. Mutation of the conserved arginine residue R3552 abrogated the binding to serine-5-phosphorylated CTD (Fig. 2c).

The SPOC domain of RBM15 interacts with the histone H3K4 methyltransferase SETD1B via an LPDSD motif similar to the SMRT/

**Fig. 1 | Conserved surfaces on SPOC mediate phosphoserine binding. a** Multiple sequence alignment of human SPOC domains based on PROMALSD3 using SPOC structures from human SHARP (2RT5), human PHF3 (6Q2V) and sequences from human RBM15, RBM15B, DIDO and SPOCD1. Colored squares indicate conserved residues that constitute basic patches on the surface of SPOC. The patches are marked in the same colors in (**b**–**f**). A red asterisk indicates an arginine residue that is conserved in all human SPOC domains except SPOCD1. Secondary structure elements are indicated above the primary sequence. **b** Crystal structure of PHF3 SPOC in complex with 2×S2P CTD peptide (6IC8). Conserved basic patches that mediate binding to phosphorylated CTD residues are indicated with green circles. The distance between the patches is 24 Å. **c** NMR solution structure of SHARP SPOC in complex with phosphorylated SMRT peptide (2RT5). The yellow circle indicates the conserved basic patch that coordinates binding to SMRT pS2522. **d** AlphaFold2 structural model DIDO SPOC (Q9BTC0). Green circles indicate conserved surface patches. The distance between the patches is 21 Å. **e** Crystal structure of RBM15 SPOC (7Z27). Colored circles indicate the conserved basic surface patch (patch 1, yellow) and a potential second patch (patch 2, orange). The distance between the patches is 21 Å. **f** AlphaFold2 structural model of SPOCD1 SPOC (Q6ZMY3). The surface patches indicated by green circles are only partially conserved and display a less pronounced positive charge. The distance between the patches is 36 Å. **g** Structural model of the interaction between RBM15 SPOC and serine-5-phosphorylated CTD generated in PyMOL and refined using the HAD-DOCK2.2 webserver[28,29]. All SPOC domains are shown in the same orientation and at the same scale. Electrostatic surface potential in (**b**–**f**) was calculated using the Coulombic Surface Coloring tool in UCSF Chimera[60] and is depicted ranging from −10 (red) to +10 (blue) kcal/(mol*e). The distances in (**b**) and (**d**–**f**) are given as the mean distance between the terminal atoms of the amino acids making up the basic patches and were measured using the structure measurements−distances tool in UCSF Chimera.

NCoR LSD motif[17]. Although it is unclear whether the serine residue in this motif is phosphorylated, K795 and K898 from the conserved basic patch are required for binding[17], which indicates that phosphoserine binding may play a role in the interaction. In our CTD-binding analysis, RBM15 SPOC bound with the highest affinity to 2×S5P CTD peptide ($K_d = 11.8 \pm 0.2\ \mu$M, Fig. 2d, e). Contrary to other SPOC domains, RBM15 SPOC also bound to 2×S2P and 2×S7P with similar affinities ($K_d = 29.6 \pm 1.4\ \mu$M for 2×S5P, $23.5 \pm 0.4\ \mu$M for 2×S7P, Fig. 2e and Supplementary Fig. 7a, b), suggesting that RBM15 SPOC might be a more promiscuous phosphoserine binding domain. We modeled RBM15 SPOC binding to an 8-mer CTD peptide phosphorylated at serine-5 (Fig. 1g). In the model, phosphorylated serine-5 is tightly coordinated by the conserved residues K795, R834, and K898 (Fig. 1g). Compared to SHARP SPOC, RBM15 SPOC had a clear preference for double phosphorylated CTD peptides (Fig. 2d, e and Supplementary Fig. 7) and mutation of the highly conserved R834 to alanine reduced, but did not completely abrogate the binding to 2×S5P CTD (Fig. 2d). This indicates that there might be a second basic patch on the surface of RBM15 SPOC. Indeed, our RBM15 SPOC structure shows a potential second interaction surface at a distance from the conserved basic patch comparable to that between the two patches of PHF3 and DIDO SPOC (Fig. 1e, patch 2, distance to patch 1: 21 Å). Amino acids R847, R848, and K850 are potential candidates for electrostatic coordination of phosphoserine from the second CTD repeat.

In summary, SPOC domains of PHF3, DIDO, SHARP, and RBM15 are CTD reader domains, which specifically recognize CTD phospho-marks via conserved basic surface patches. While PHF3, DIDO, and RBM15 SPOC are tuned for binding double phosphorylated CTD motifs, SHARP SPOC recognizes a single CTD phosphorylation.

**Acidic residues determine the binding affinity of SHARP SPOC**

SHARP SPOC was reported to bind to the phosphorylated LSD motif of the SMRT/NCoR C-terminus with nanomolar affinity determined by SPR and ITC[3,11]. To allow a direct comparison with the binding affinity to serine-5-phosphorylated CTD, we performed fluorescence aniso-tropy using FAM-labeled NCoR peptides that were either unphosphorylated or phosphorylated at the CK2 site S2436, which was previously shown to be critical for SHARP SPOC binding[11] (Fig. 3a). SHARP SPOC bound to phosphorylated NCoR with an affinity of $1.75 \pm 0.04\ \mu$M. Unphosphorylated NCoR had a reduced binding affinity ($K_d = 13.6 \pm 1.3\ \mu$M), but in contrast to previously published data, the binding was not completely lost[11]. This discrepancy may be due to different experimental setups used to determine binding affinities. In line with our results, SHARP SPOC was shown to bind to the unphosphorylated C-terminus of SMRT, albeit with lower affinity than to its phosphorylated counterpart[3].

SHARP SPOC binds to SMRT/NCoR with substantially higher affinity compared to the CTD ($K_d = 1.75 \pm 0.04\ \mu$M and $K_d = 23.8 \pm 0.8\ \mu$M, respectively) (Figs. 2c and 3a). To define the molecular determinants for the preferential binding to SMRT/NCoR, we solved the crystal structure of SHARP SPOC in complex with 1×S5P CTD peptide at a resolution of 1.55 Å (Fig. 3b, d, Supplementary Table 1). pS5 of the CTD is electro-statically anchored to the conserved basic surface patch of SHARP SPOC (Fig. 3b). The NH1 nitrogen of R3552 and the ε-aminogroups of K3516 and K3606 form hydrogen bonds with O1P, O2P and O3P of CTD pS5 (Fig. 3d). The binding mode of SHARP SPOC to 1×S5P CTD is remarkably similar to the binding between SHARP SPOC and the C-terminus of SMRT (Figs. 1c and 3b–e). Both peptides occupy the same surface on the SPOC domain. pS5 of the CTD and pS2522 of SMRT are anchored via electrostatic interactions with K3516, R3552, and K3606 (Fig. 3d, e). A notable difference between the two structures is elec-trostatic interactions between the SMRT peptide and R3548 of SHARP SPOC, which is coordinated by D2523 and E2525 of SMRT (Fig. 3e, dashed red circle). These acidic residues are also present at the same position in the NCoR C-terminus (SMRT: -YETLp**SDSE**, NCoR: -YETLp**SDSD**D), but are absent from the CTD (-YSPTpSPSYSPTSPS) impeding hydrogen bonding with R3548 (Fig. 3d). The additional electrostatic interactions confer strong binding to SMRT even in the absence of serine phosphorylation and might explain why SHARP SPOC exhibits higher affinity for SMRT/NCoR compared to the CTD.

Tyrosine residues play an important role in phosphoserine recognition and determining the register of CTD-phosphoserine recognition. The conserved residue Y1291 in PHF3 SPOC/Y3602 in SHARP SPOC is involved in coordinating pS2 of the second CTD repeat in the case of PHF3 or pS5 in the case of SHARP SPOC (Fig. 3d, f). In addition, PHF3 residues Y1257 and Y1312, together with T1253, form a hydrophobic pocket for P6 of the first CTD repeat (Fig. 3f). These hydrophobic contacts mediate tight packing of the CTD peptide to the PHF3 SPOC surface and specific recognition of tandem S2 phosphorylation. Y1257 and Y1312 are not conserved in SHARP, which may explain the different modes of binding and why SHARP SPOC recognizes a single rather than two adjacent phosphorylated CTD repeats.

Given the comparable spatial proximity of SMRT residues pS2522, D2523, and E2525 and CTD residues S5 and S7, we hypothesized that double phosphorylation of CTD at serine-5 and serine-7 might mimic the negative charge of SMRT D2523 and E2525 and confer stronger binding of SHARP SPOC to CTD. To explore this, we used HADDOCK 2.2[28,29] to model the binding of SHARP SPOC to 8-mer SMRT pS2522, CTD pS5, and CTD pS5pS7 peptides (Fig. 3g). Based on the HADDOCK score, the strongest interaction was predicted between SHARP SPOC and phosphorylated SMRT. The predicted binding strength for double phosphorylated pS5pS7 CTD was similar to that for single phos-phorylated pS5CTD, indicating that pS7 cannot compensate for the absence of acidic residues in the CTD. Similarly, RBM15 SPOC was predicted to bind CTD pS5 and CTD pS5pS7 peptides with comparable affinity (Fig. 3g).

Collectively, our findings establish SHARP SPOC as a phospho-serine binding module that preferentially binds phosphorylated LSD

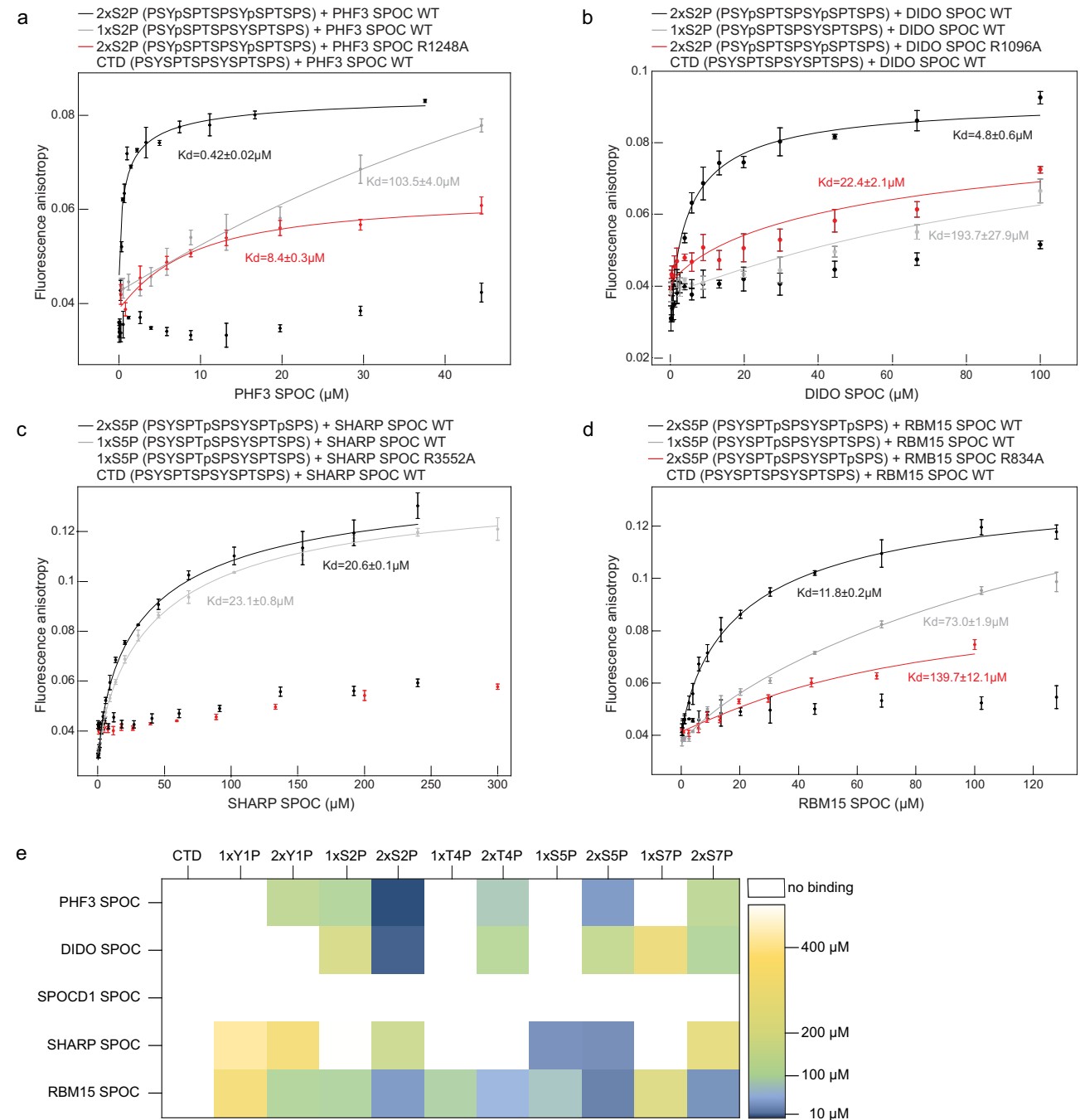

**Fig. 2 | SPOC is a CTD binding domain. a–d** Fluorescence anisotropy measurements of SPOC domains and CTD peptides. Fluorescence anisotropy is plotted as a function of protein concentration. Data points and error bars represent the mean ± standard deviation from three independent experiments. **e** Heatmap of binding affinities of SPOC domains to CTD peptides. Source data are provided as a Source Data file.

motifs with adjacent acidic residues but can accommodate different phosphorylated binding partners of SHARP via its conserved basic patch.

**Transcription machinery is the major anchoring point for PHF3 and DIDO SPOC domains**

To further elucidate how the SPOC domain shapes the interaction network of SPOC-containing proteins, we performed co-immunoprecipitation (co-IP) of FLAG-tagged PHF3, DIDO, SHARP, and RBM15 SPOC domains expressed in HEK293T cells and identified their interaction partners by mass spectrometry (Fig. 4 and Supplementary Data 1). The interactions were confirmed by

Western blotting (Fig. 5a). Moreover, to examine SPOC-mediated interactions in the context of full-length proteins, we expressed FLAG-tagged full-length (wt) and SPOC-deleted (ΔSPOC) PHF3, DIDO, SHARP, and RBM15 in HEK293T cells and performed anti-FLAG co-IP (Fig. 5b–e). All SPOC domains and full-length proteins interacted with Pol II. DIDO and PHF3 SPOC showed stronger binding compared to SHARP and RBM15, in accordance with in vitro binding assays (Figs. 5a and 2). The association of PHF3 SPOC with Pol II appeared weaker compared to DIDO, which may be due to the lower expression level of FLAG-tagged PHF3 SPOC (Fig. 5a). Binding of PHF3, DIDO, and RBM15 to Pol II was abrogated upon loss of the SPOC domain, indicating that the SPOC–CTD

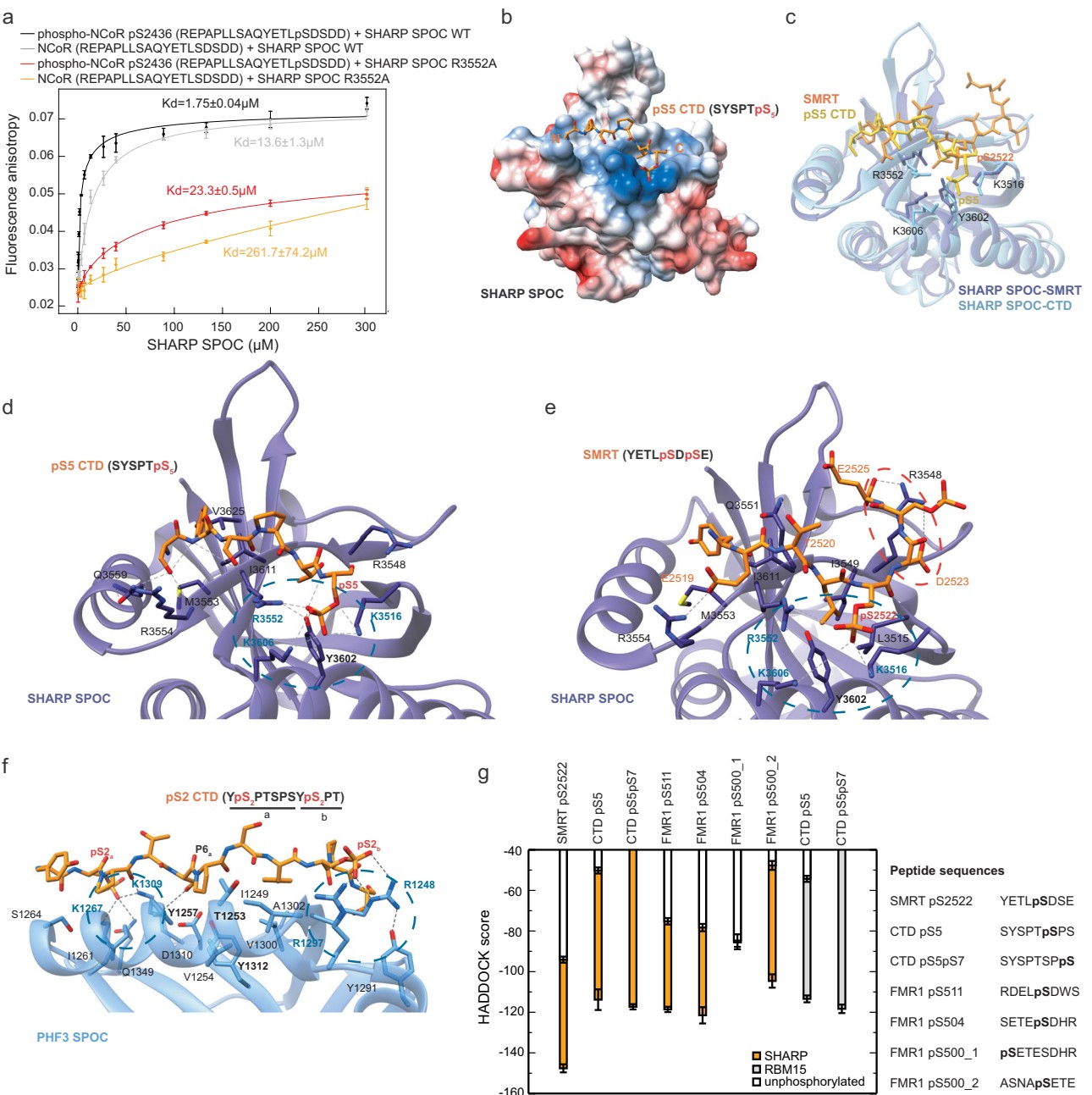

**Fig. 3 | Acidic residues determine SHARP SPOC binding affinity. a** Fluorescence anisotropy binding curves of SHARP SPOC with NCoR peptides. Fluorescence anisotropy is plotted as a function of protein concentration. Data points and error bars represent the mean ± standard deviation from three independent experiments. Source data are provided as a Source Data file. **b** Crystal structure of SHARP SPOC in complex with 1×S5P CTD peptide (7Z1K). Electrostatic surface potential was calculated using the Coulombic Surface Coloring tool in UCSF Chimera and is depicted ranging from −10 (red) to +10 (blue) kcal/(mol*e). **c** Structural comparison of SHARP SPOC binding to 1×S5P CTD and SMRT (2RT5). **d** Interactions between SHARP SPOC and 1×S5P CTD. Hydrogen bonds are indicated with dashed lines. The blue circle indicates a conserved basic surface patch. **e** Interactions between SHARP SPOC and phosphorylated SMRT (2RT5). Hydrogen bonds are indicated with dashed lines. The blue circle indicates the conserved basic surface patch, red circle indicates additional electrostatic coordination of R3548 of SPOC by D2523 and E2525 of SMRT, which is not possible with CTD. **f** Interactions between PHF3 SPOC and 2×S2P CTD (6Q2V). Hydrogen bonds are indicated with dashed lines. Blue circles indicate conserved basic surface patches. **g** HADDOCK scores of modeled interactions between SHARP SPOC or RBM15 SPOC and SMRT, CTD, or FMR1 peptides. Peptides used for modeling were either unphosphorylated (white bars) or phosphorylated at the indicated residues (SHARP SPOC: orange bars, RBM15 SPOC: gray bars). Bars correspond to the HADDOCK score ± SD calculated by HADDOCK. Source data are provided as a Source Data file.

interaction is of critical importance in establishing their interaction with Pol II (Fig. 5b, c, e). However, SPOC deletion in SHARP did not impair interaction with Pol II, suggesting that it contacts Pol II via multiple surfaces (Fig. 5d).

In addition to Pol II, PHF3 and DIDO showed SPOC-dependent interaction with transcription elongation factors such as SPT5, SPT6/ IWS1 and the PAF1 complex (PAF1, LEO1, CTR9, CDC73, WDR61)

(Figs. 4a, b, e and 5a–c). Interestingly, all SPOC domains interacted with the PAF1 complex, which was not the case for full-length RBM15 and SHARP (Fig. 5a, d, e). PHF3 and DIDO interaction with the transcription factor ZNF768, which contains a heptad repeat sequence structurally related to Pol II CTD[30] was not abrogated upon SPOC deletion (Fig. 5b, c). In contrast to PHF3, full-length DIDO and its SPOC domain showed interaction with CK2 (casein kinase 2), which was slightly diminished

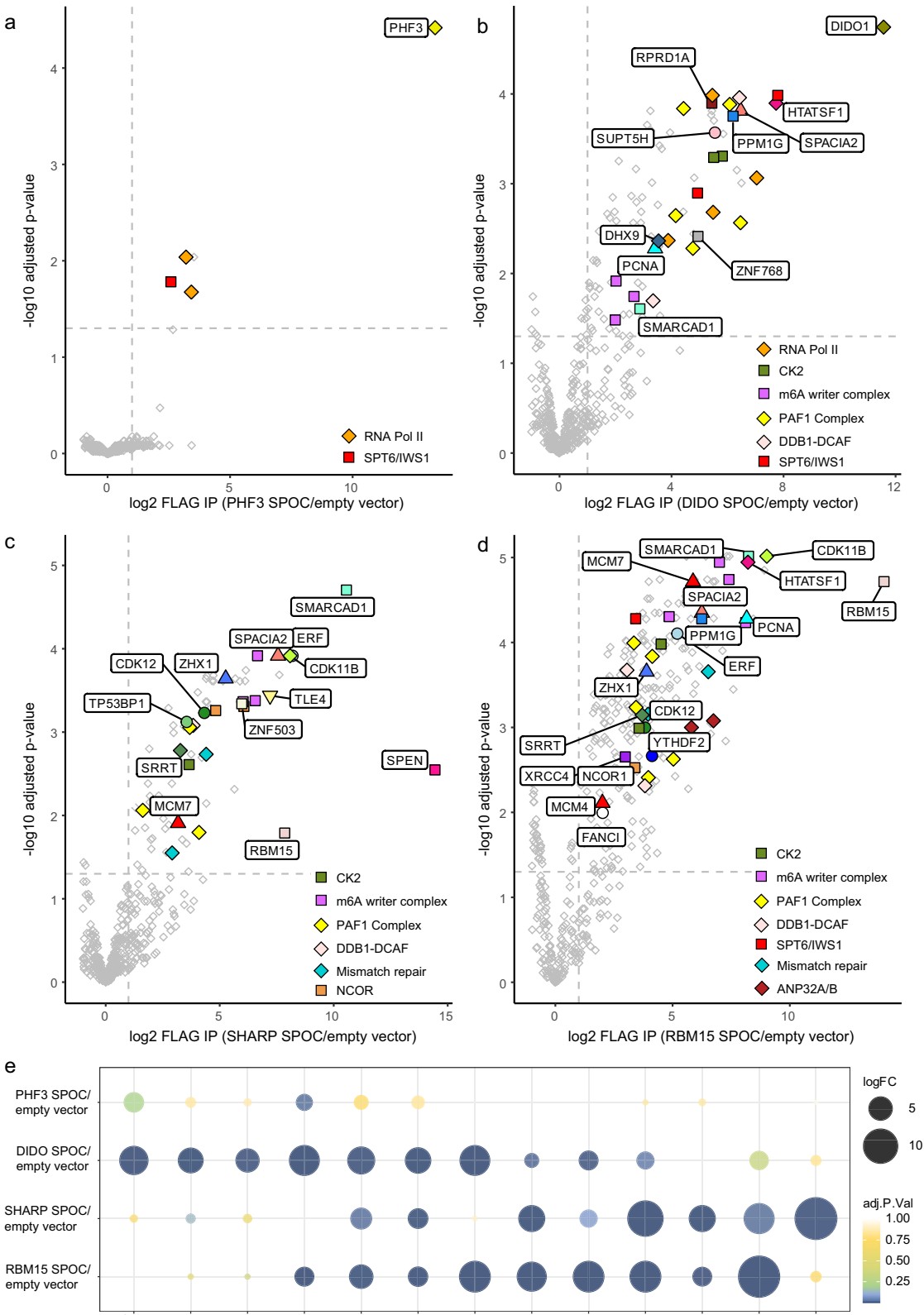

**Fig. 4 | Mass spectrometry analysis of SPOC domain interactome. a–d** Volcano plots of **a** PHF3, **b** DIDO, **c** SHARP, **d** RBM15 SPOC interactors identified by mass spectrometry. **e** Overview of common SPOC domain interactors identified by mass spectrometry. The experiments were performed in three individual replicates. Statistical tests were performed using the LIMMA package[65].

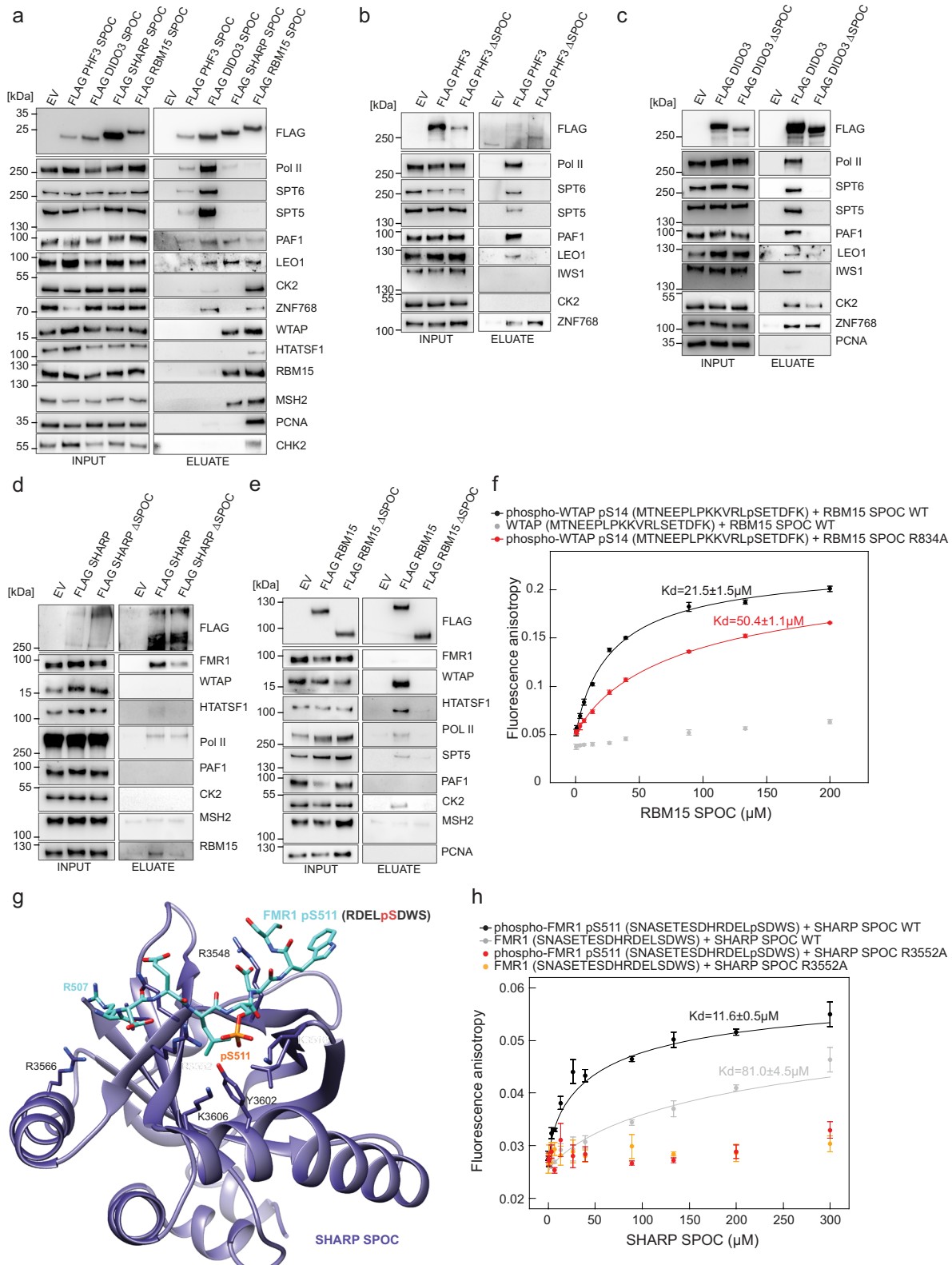

**Fig. 5 | Interactome of SPOC proteins. a** Western Blot analysis after FLAG-co-IP of FLAG-tagged SPOC domains. **b**–**e** Western Blot analysis after FLAG-co-IP of FLAG-tagged full-length and ΔSPOC proteins. The experiments in **a**–**e** were performed once. **f** Fluorescence anisotropy binding curves of RBM15 SPOC with WTAP peptides. Fluorescence anisotropy is plotted as a function of protein concentration. Data points and error bars show mean anisotropy ± standard deviation from three independent experiments. **g** Structural model of the interaction between SHARP SPOC and an 8-mer FMR peptide phosphorylated at S511 generated in PyMOL and refined using the HADDOCK2.2 webserver[28,29]. Repulsion between SHARP R3566 and FMR R507 may lead to reduced binding affinity compared to SMRT. **h** Fluorescence anisotropy binding curves of SHARP SPOC with FMR1 peptides. Fluorescence anisotropy is plotted as a function of protein concentration. Data points and error bars show mean anisotropy ± standard deviation from three independent experiments. Source data are provided as a Source Data file.

upon SPOC deletion (Fig. 5a, c). CK2 is known to phosphorylate a number of transcription factors, including the LSD motif of the cor-epressor SMRT[3,31,32]. The interaction between RBM15 SPOC and CK2 could not be recapitulated with the full-length RBM15 (Fig. 5a, e).

Taken together, our results establish the SPOC domain as an essential module for PHF3 and DIDO interaction with the Pol II tran-scription elongation machinery, whereas SHARP and RBM15 SPOC show stronger binding to other proteins.

## SPOC domain mediates RBM15 and SHARP interaction with the m⁶A regulators

Major interactors of the RBM15 SPOC domain comprised the m$^6$A writer complex components WTAP, ZC3H13, and VIRMA/KIAA1429, the mismatch repair protein MSH2, the replication scaffold protein PCNA, as well as HTATSF1 (TAT-SF1), transcription elongation and splicing factor that couples transcription with pre-mRNA processing[33] (Figs. 4d and 5a). Full-length RBM15 interacted strongly with the m$^6$A complex member WTAP and HTATSF1 in a SPOC-dependent manner but not with MSH2 or PCNA (Fig. 5e).

RBM15 is an established component of the m$^6$A writer complex, however, the exact nature of the interaction with the other subunits and how the interaction is regulated remains unknown. Given that the isolated SPOC domain strongly interacts with WTAP (Figs. 4d and 5a) and the interaction of full-length RBM15 with WTAP is SPOC-dependent (Fig. 5e), we hypothesized that a phosphorylation-dependent SPOC-WTAP contact may be the primary anchoring point for the interaction of RBM15 and the m$^6$A writer complex. Using PhosphoSite Plus[34] we identified an LSETD motif reminiscent of the SETD1B LPDSD motif bound by RBM15 SPOC close to the N-terminus of WTAP. This motif includes the phosphorylated residue serine-14.

We determined the binding affinity of RBM15 SPOC to a peptide corresponding to the first 19 N-terminal amino acids of WTAP (Fig. 5f). RBM15 SPOC bound to the peptide phosphorylated at S14 with an affinity of $21.5 \pm 1.5\,\mu M$ but did not bind to the unphosphorylated peptide. Mutation of the conserved residue R834 to alanine reduced the binding affinity to $50.4 \pm 1.1\,\mu M$ but did not abrogate the binding, indicating that additional residues in RBM15 SPOC contribute to WTAP pS14 recognition. ZC3H13 was suggested to act as a bridge between RBM15 and WTAP to recruit the METTL3/14 complex[16]. However, we did not detect binding between RBM15 SPOC and phosphorylated or unmodified ZC3H13 peptides (Supplementary Fig. 7e).

In contrast to RBM15, the interaction of SHARP SPOC with WTAP did not translate to the full-length protein (Fig. 5d). Considering the limitations of using protein overexpression and the lack of SHARP antibodies that would allow immunoprecipitation of the endogenous protein, we decided to tag SHARP and SHARP ΔSPOC with GFP at the C-terminus using CRISPR/Cas9 knock-in approach and perform anti-GFP co-IP followed by mass spectrometry (Supplementary Figs. 8a, c and 9, Supplementary Data 2). This analysis showed SPOC-dependent interaction of SHARP with FMR1, FXR1, and FXR2, which was confirmed for FMR1 by Western blotting (Fig. 5d and Supplementary Data 2). FMR1, also called FMRP (fragile X mental retardation protein), binds to m$^6$A mRNA and mediates its export into the cytoplasm, which is essential for neuronal differentiation[35]. Both FXR1 and FXR2 (fragile X-related proteins 1 and 2) regulate adult neurogenesis and have been implicated in various neurological disorders, but the underlying mechanisms are not well understood[36]. Given that SHARP SPOC binds to LSD motifs at the C-terminus of SMRT/NCoR[3,11], we hypothesized that SHARP may contact FMR1 via a similar motif. We identified an LSD motif with several phosphorylated residues based on PhosphoSitePlus[34] and used HADDOCK 2.2[28,29] to refine SHARP/FMR1 complexes, which were reconstructed using the SHARP-SMRT structure (2RT5) as a template (Fig. 3g). According to the modeling, SHARP interactions with FMR1 are weaker compared to SMRT. This can be due to the absence of a tyrosine residue at the

corresponding position (Y2518 in SMRT or Y1 in CTD) that interacts with SHARP R3566 via π-π interactions and forms hydrophobic con-tacts with M3553. In the FMR1 pS511 peptide, the residue in this posi-tion is arginine (R507) instead of tyrosine which can lead to repulsive interaction with R3566 of SHARP SPOC (Fig. 5g).

We determined the binding affinity of SHARP SPOC to the FMR1 peptide surrounding pS511 by fluorescence anisotropy (Fig. 5h). SHARP SPOC bound to the FMR1 pS511 peptide with an affinity of $11.6 \pm 0.5\,\mu M$, further showing how the SHARP SPOC domain med-iates interaction with a variety of phosphoserine binding partners. Affinity for the unphosphorylated peptide was strongly reduced ($K_d = 81.0 \pm 4.5\,\mu M$), while mutation of the conserved R3552 to ala-nine abrogated the binding to both peptides (Fig. 5h).

Overall, our results suggest that the SPOC domains of the Spen family members RBM15 and SHARP mediate interactions with the m$^6$A writer and reader proteins.

## SPOC proteins regulate pause release, mRNA stability, and m6A abundance

To probe the cellular function of the SPOC domain in the context of gene expression regulation, we generated DIDO KO and DIDO ΔSPOC, RBM15 and RBM15 ΔSPOC, as well as SHARP ΔSPOC-GFP HEK293T cell lines using CRISPR/Cas9 (Supplementary Figs. 8a, c, 10, and 11). We could not generate SHARP KO cells, suggesting that this gene is essential for the viability of HEK293T cells. PHF3 KO and PHF3 ΔSPOC were previously generated[5].

To determine whether loss of SPOC domain proteins or the SPOC domain leads to perturbations in transcription or altered mRNA levels, we performed RNA-seq and TT$_{chem}$-seq[37] to analyze the levels of mature RNA and nascent transcripts, respectively (Figs. 6 and 7 and Supplementary Data 3 and 4). We used MA plots to show expression changes in KO or ΔSPOC cell lines relative to WT (Fig. 6 and Supple-mentary Data 3), box plots showing the genome-wide distribution of log2 fold changes in TT$_{chem}$-seq signal in the KO and ΔSPOC cell lines relative to WT to visualize alterations in nascent transcript levels at TSS (transcription start sites) and gene bodies (Fig. 7a, b and Supplemen-tary Data 4), we determined the stalling index as TT$_{chem}$-seq TSS/gene body reads (Fig. 7c and Supplementary Data 4), and we calculated differences in log2 fold changes KO/WT or ΔSPOC/WT between RNA-seq and TT$_{chem}$-seq to assess changes in mRNA stability (Fig. 7d).

For PHF3, we used our previously published RNA-seq dataset[5], which showed the upregulation of mature transcripts in PHF3 KO or ΔSPOC cells relative to WT (Fig. 6a–c). TT$_{chem}$-seq showed reduced nascent transcript levels and slightly increased Pol II stalling in PHF3 KO and ΔSPOC cells (Fig. 7a–c), indicating that PHF3 positively reg-ulates transcription. Differences between RNA-seq and TT$_{chem}$-seq pointed to increased mRNA stability (Fig. 7d), as previously shown[5]. DIDO KO and ΔSPOC cells showed downregulation of >900 tran-scripts, slightly reduced Pol II stalling, and reduced mRNA stability (Figs. 6d–f and 7), suggesting that DIDO negatively regulates tran-scription but positively regulates mRNA stability. Neuronal genes were enriched among upregulated genes in PHF3 KO/ΔSPOC and down-regulated genes in DIDO KO/ΔSPOC (Supplementary Fig. 12). Inter-estingly, in SHARP ΔSPOC cells, nascent transcript levels were equally reduced at TSS and gene body regions, without a change in stalling index (Fig. 7a–c) and 165 mature transcripts were repressed in SHARP ΔSPOC without a change in mRNA stability (Figs. 6i and 7d), suggesting that SHARP through its SPOC domain positively regulates transcrip-tion initiation. RBM15 KO and ΔSPOC gave rise to a major repression of mature transcripts (16674 in RBM15 KO and 14281 in RBM15 ΔSPOC) (Fig. 6g, h, j), increased stalling (Fig. 7c), and a decrease in mRNA stability (Fig. 7d), indicating that RBM15 positively regulates mRNA stability through its SPOC domain.

Since RBM15 and SHARP interact with writers and readers of the m$^6$A RNA modification via their SPOC domains, we investigated

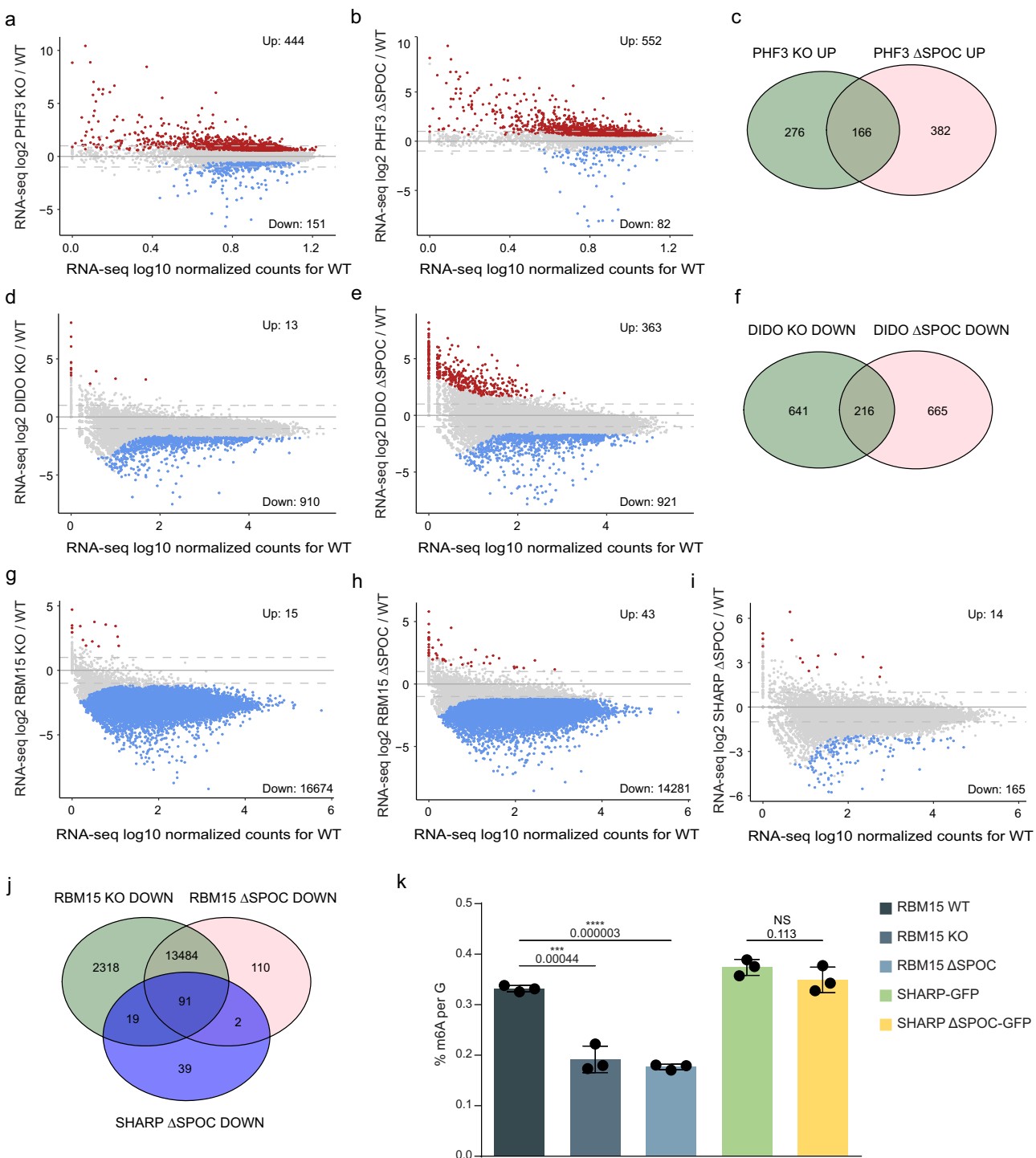

**Fig. 6 | SPOC domain proteins regulate gene expression and m⁶A levels.**
**a**, **b**, **d**, **e**, **g**–**i** MA plots showing RNA-seq log2 fold change (KO/WT or ΔSPOC/WT) versus log10 mean expression in WT for **a** PKF3 KO, **b** PHF3 ΔSPOC, **d** DIDO KO, **e** DIDO ΔSPOC, **g** RBM15 KO, **h** RBM15 ΔSPOC, and **i** SHARP ΔSPOC. Red and blue dots indicate upregulated and downregulated genes, respectively with fold-change > 2, p < 0.05. Statistical analysis was performed using the Wald test as implemented in DESeq2[72]. Drosophila S2 cells were used for spike-in normalization. **c**, **f**, **j** Venn diagram showing overlaps between **c** upregulated genes in PHF3 KO and

PHF3 ΔSPOC, **f** downregulated genes in DIDO KO and DIDO ΔSPOC, **j** downregulated genes in SHARP ΔSPOC, RBM15, and RBM15 ΔSPOC. **k** m⁶A levels are decreased upon impairment of RBM15. Mass spectrometry analysis of single nucleosides derived from mRNA isolated from the indicated cell lines. Data are presented as mean ± standard deviation of three replicates, and individual data points are indicated as black dots. One-tailed, two-sample equal variance t-test was used to determine p-values. Source data are provided as a Source Data file.

whether the loss of these proteins or their SPOC domains results in changes in cellular m⁶A levels. To this end, we isolated mRNA from RBM15 WT, KO and ΔSPOC, and SHARP-GFP and ΔSPOC-GFP and measured the relative abundance of individual nucleosides by mass spectrometry. The analysis revealed a prominent decrease of

m⁶A modification in mRNA from RBM15 KO and RBM15 ΔSPOC, whereas SPOC deletion in SHARP did not have a pronounced effect (Fig. 6k). This indicates that RBM15 and its SPOC domain are critical for m⁶A modification of mRNA, while SHARP is dispensable, which is not surprising given that it interacts with the m⁶A reader

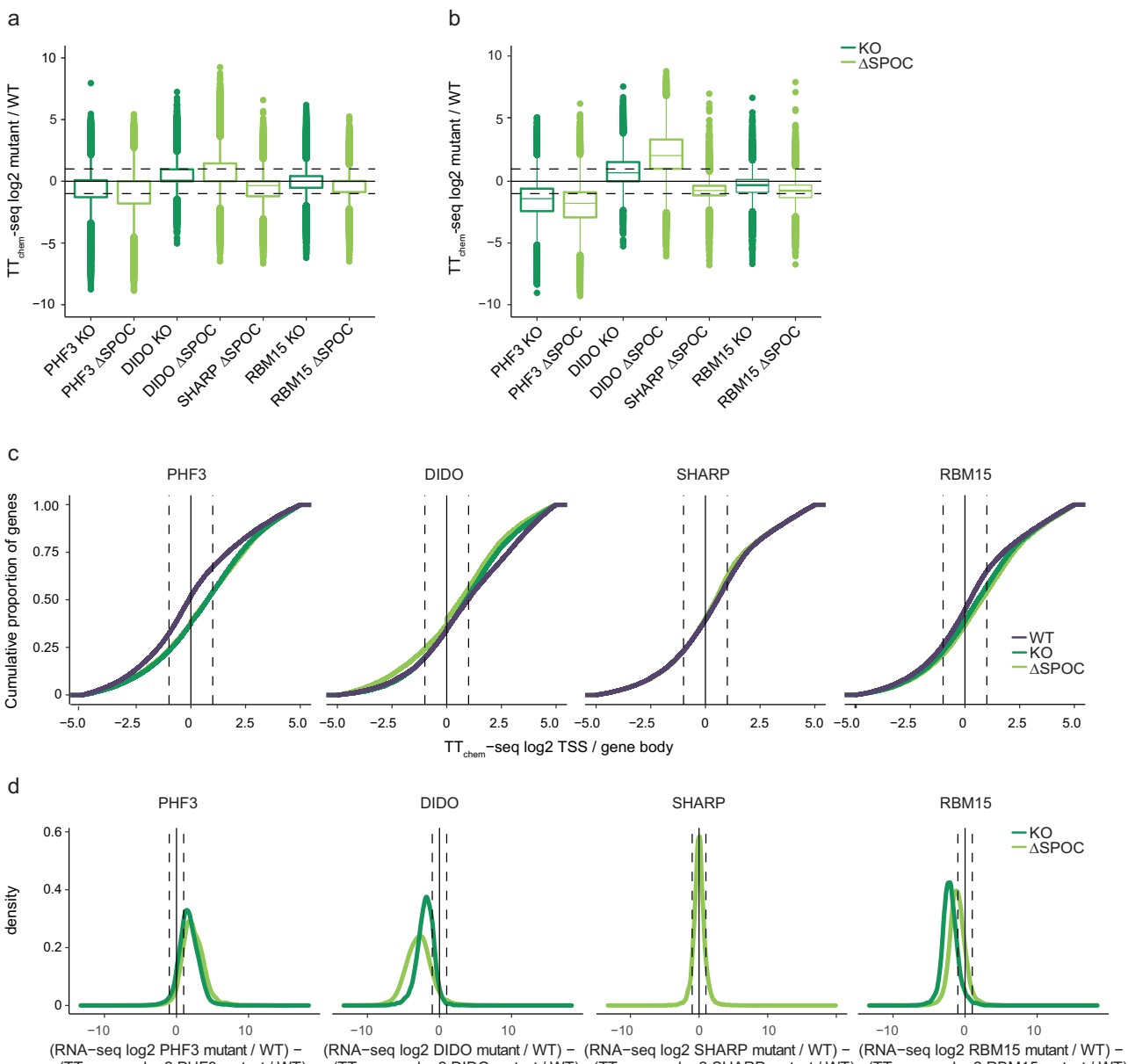

**Fig. 7 | SPOC domain proteins regulate transcription and mRNA stability.**
**a**, **b** Genome-wide distribution of log2 fold changes of TT_chem-seq signal in KO and
ΔSPOC relative to WT on **a** TSS and **b** gene body regions. Box plots show the median
(central line), the 25–75% interquartile range (IQR) (box limits), and the ±1.5× IQR
(whiskers). Median, minimum, maximum, and 25 and 75% percentile values are
provided in the Source Data file. Cells were treated with 1 mM 4sU for 15 min. In

vitro transcribed synthetic 4sU-labeled RNA and 4sU-labeled yeast RNA were used
for spike-in normalization. Experiments were performed in three independent
replicates. **c** Stalling index analysis calculated as TT_chem-seq TSS/gene body signal.
**d** Density distribution of the differences in log2 fold changes KO/WT or ΔSPOC/WT
between RNA-seq and TT_chem-seq data. Source data are provided as a Source
Data file.

FMR1 rather than the m⁶A writer complex and is thus more likely to
influence the fate of m⁶A-modified RNAs rather than the estab-
lishment of the modification itself.

## SPOC determines the genomic localization of PHF3 and SHARP

SPOC proteins regulate transcription and co-transcriptional pro-
cesses. Thus, their recruitment to chromatin is crucial for their
proper function. We performed immunofluorescence microscopy
to determine whether loss of the SPOC domain affects cellular
localization of the respective protein (Fig. 8a). PHF3, DIDO, and
RBM15 showed no major changes in localization upon SPOC dele-
tion. Conversely, SHARP was enriched in two clusters that are
reminiscent of Barr bodies, in which two out of three copies of the

X-chromosome in HEK293T are inactivated by Xist long non-coding
RNA. Strikingly, SPOC deletion led to the complete delocalization of
SHARP (Fig. 8a).

SHARP mediates X-chromosome inactivation by binding to Xist
lncRNA[12]. To verify the importance of the SPOC domain for binding
to the *XIST* locus, we employed Chromatin Immunoprecipitation
followed by high-throughput sequencing (ChIP-seq) analysis
(Fig. 8b). Due to a lack of suitable antibodies for immunoprecipi-
tation of the endogenous proteins, we performed GFP-ChIP on
endogenously tagged SHARP-GFP and SHARP ΔSPOC-GFP cell
lines (Supplementary Fig. 8a, c). ChIP-seq analysis of SHARP-GFP
revealed only one area of enriched genomic binding at the *XIST*
locus in HEK293T cells, which we confirmed by quantitative
polymerase chain reaction (qPCR) using multiple primer pairs

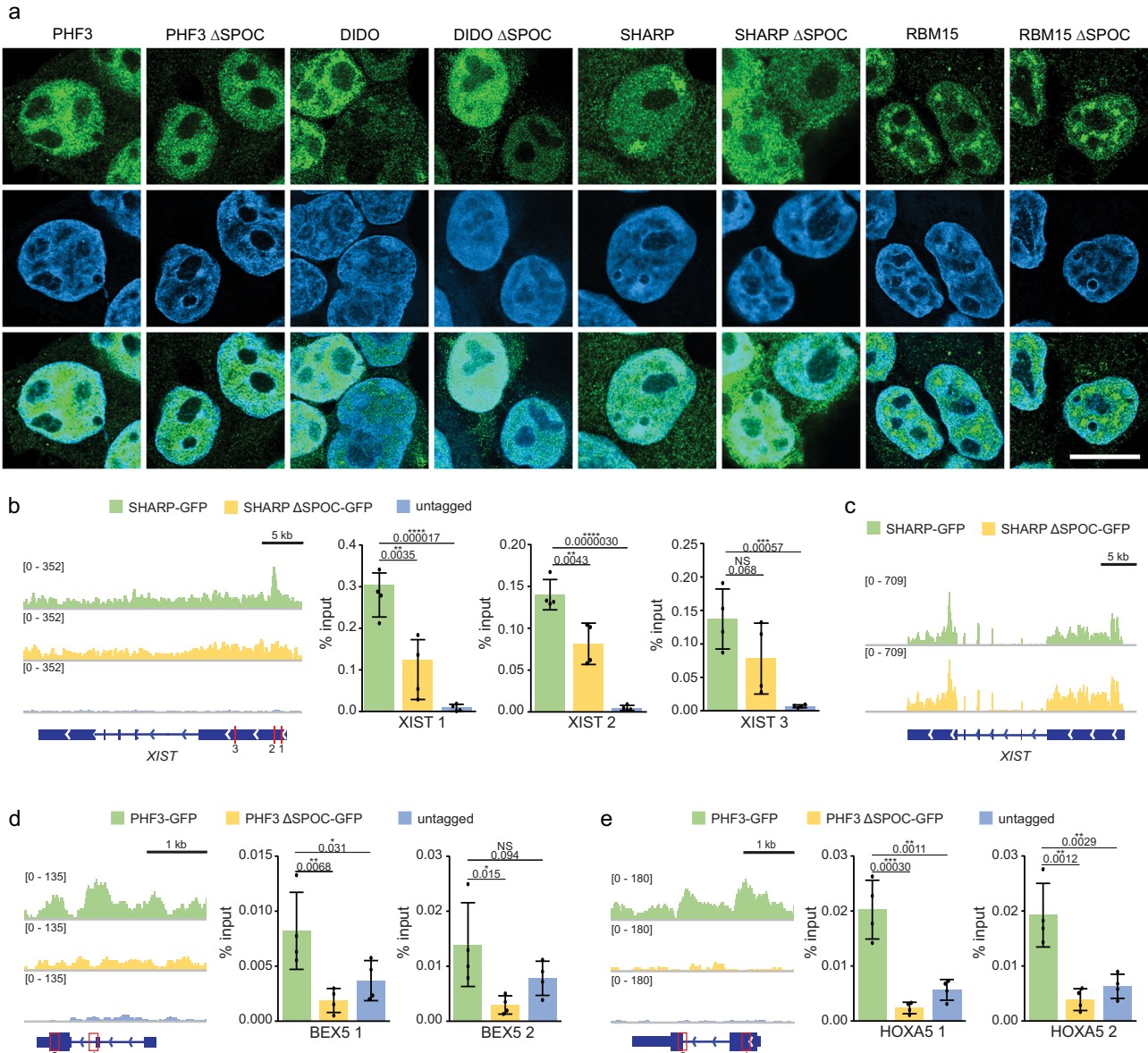

**Fig. 8 | SPOC domains determine genomic localization. a** Cellular localization of SPOC proteins analyzed by immunofluorescence microscopy. Exemplary Airyscan images showing the localization of wild-type and SPOC-deleted proteins for PHF3, DIDO, SHARP, and RBM15 (green). DAPI and merged images are shown below. Localization of each protein of interest was acquired in biological duplicates. Scale bar = 10 μm. **b** Integrative Genomics Viewer (IGV) snapshots showing GFP-ChIP-seq reads and GFP-ChIP-qPCR analysis for XIST in SHARP-GFP, SHARP ΔSPOC-GFP, and untagged HEK293T cells. **c** IGV snapshots showing RNA-seq reads for XIST in SHARP-GFP and SHARP ΔSPOC-GFP cells. **d** IGV snapshots showing GFP-ChIP-seq reads and GFP-ChIP-qPCR analysis for BEX5 in PHF3-GFP, PHF3 ΔSPOC-GFP, and untagged HEK293T cells. **e** IGV snapshots showing GFP-ChIP-seq reads and GFP-ChIP-qPCR analysis for HOXA5 in PHF3-GFP, PHF3 ΔSPOC-GFP, and untagged HEK293T cells. The qPCR data in **b**, **d**, **e** are presented as mean ± standard deviation of four individual experiments, individual data points are indicated as black dots. One-tailed, two-sample equal variance *t*-test was used to determine *p*-values. qPCR amplicons are indicated as red boxes. Source data are provided as a Source Data file.

targeting the *XIST* locus (Fig. 8b). SHARP ΔSPOC was less strongly recruited to chromatin than the full-length protein (Fig. 8b), corroborating SHARP ΔSPOC delocalization from the Barr bodies observed by immunofluorescence microscopy (Fig. 8a). However, *Xist* expression levels were not reduced in SHARP ΔSPOC (Fig. 8c).

We had previously generated PHF3-GFP and PHF3 ΔSPOC cell lines[5] and applied CRISPR/Cas9 knock-in to tag PHF3 ΔSPOC with GFP at the C-terminus (Supplementary Fig. 8b, d). ChIP-seq analysis revealed that PHF3-GFP bound to expressed genes and that recruitment to chromatin was reduced upon deletion of the SPOC domain, indicating that the interaction with phosphorylated Pol II CTD is

critical for PHF3 binding to chromatin (Supplementary Fig. 13a and Supplementary Data 5). Moreover, genes that showed reduced chromatin association were deregulated in PHF3 ΔSPOC according to RNA-seq (Supplementary Fig. 13b). Among PHF3 target genes were *BEX5* and *HOXA5* (Fig. 8d, e), We confirmed the binding of PHF3-GFP to these genomic loci by ChIP-qPCR (Fig. 8d, e). Both in sequencing and qPCR analysis, PHF3 recruitment was strongly reduced upon deletion of the SPOC domain.

Our results indicate that SHARP and PHF3 are recruited to chromatin via the SPOC domain, further highlighting the important role of this domain in SPOC protein-mediated gene regulation.

## Discussion

We established SPOC as a CTD reader domain that recognizes different CTD phosphorylation patterns. PHF3 and DIDO SPOC bind to CTD phosphorylated at serine-2, while SHARP SPOC binds to phosphorylated serine-5. RBM15 SPOC has a preference for serine-5 phosphorylation but also binds phosphorylated serine-2 and serine-7 with similar affinity. Serine-5 and serine-7 phosphorylation are prominent in the early stages of transcription and decrease during the elongation phase concurrent with an increase of serine-2 phosphomarks[18]. This suggests that SHARP binds to Pol II during early transcription, PHF3 and DIDO regulate the elongation stage, while RBM15 might function in different phases of transcription (Fig. 9). Indeed, we found that SHARP positively regulates transcription initiation through its SPOC domain. Conversely, PHF3, DIDO, and RBM15 regulate transcription elongation and mRNA stability. PHF3 and RBM15 positively regulate pause release, whereas DIDO negatively regulates pause release. PHF3 negatively regulates mRNA stability, whereas DIDO and RBM15 have a positive effect on mRNA stability.

DIDO SPOC, like the SPOC domain of its paralogue PHF3, preferentially binds to CTD phosphorylated on serine-2 in two adjacent repeats. The SPOC domain is essential for interaction with Pol II. Interestingly, the SPOC domain is only present in the longest isoform, DIDO3; the short isoform DIDO1 does not contain SPOC, while the DIDO2 SPOC domain is truncated and likely unfunctional. Since association with Pol II and the elongation complex is dependent on the SPOC domain, direct interaction of DIDO1 and DIDO2 with Pol II is improbable. DIDO3 is the dominant isoform in embryonic stem cells and promotes maintenance of the stem cell state; a switch to the expression of the short isoform DIDO1 triggers differentiation[22]. While DIDO3 is expected to regulate transcription through direct interaction with Pol II, the shorter isoforms may indirectly modulate gene expression through chromatin binding.

In vitro binding assays showed that DIDO SPOC binds to 2×S2P CTD with a tenfold lower affinity compared to PHF3 SPOC (Fig. 2a, b). However, co-IP experiments with SPOC domains or full-length proteins show that DIDO may have a similar or even stronger affinity for Pol II in cells, which may be due to higher expression levels and stability compared to PHF3 (Fig. 5a–c). While DIDO and PHF3 dock onto the Pol II elongation complex through interaction with CTD phosphorylated at serine-2, they can establish additional contacts with Pol II, for instance, through their TLD domain, which has been shown to bind to the Pol II jaw domain in their yeast homolog Bye1[38]. The TLD shares homology with the central domain of transcription factor IIS but lacks the ability to rescue backtracked Pol II by stimulating RNA cleavage, which is conferred by TFIIS domain III[39]. PHF3 can outcompete TFIIS for Pol II binding and thereby repress transcription[5]. It remains to be addressed in future studies if DIDO regulates transcription in a similar fashion and whether PHF3 and DIDO share regulatory functions. Given that both PHF3 and DIDO bind to the same CTD phosphomark, it is conceivable that they might either bind to the elongation complex simultaneously and exercise synergistic functions or that their binding might be mutually exclusive and they compete for Pol II binding. Given their important roles in neuronal development[5,9], they might also act redundantly. While genomic localization of PHF3 is largely dependent on the SPOC domain, DIDO PHD, in contrast to PHF3 PHD, binds to H3K4me3 histone marks and can thus bind chromatin independently of SPOC[40,41]. DIDO may therefore have independent functions that cannot be taken over by PHF3.

It was previously shown that SHARP SPOC interacts with both SMRT/NCoR-containing corepressor complexes and KMT2D-containing activator complexes, with phosphorylation of the SMRT/NCoR LSD motif shifting the balance to the repressive complex[11]. Our study reveals additional members of the SHARP SPOC interactome. We showed that SHARP SPOC binds to Pol II CTD phosphorylated on serine-5, albeit with lower affinity than to SMRT/NCoR. We solved the structure of SHARP SPOC in complex with the CTD, which revealed that the strength of binding depends in part on an acidic residue adjacent to the LSD motif of SMRT/NCoR, which is missing in the CTD. Although co-IP experiments showed that the SPOC domain is not essential for the interaction between SHARP and Pol II (Fig. 5d), ChIP analysis revealed reduced recruitment of SHARP ΔSPOC onto the Xist locus, while immunofluorescence analysis revealed its delocalization from inactive X-chromosomes (Fig. 8a, b). This suggests bimodal recruitment of SHARP onto the Xist locus on the X-chromosome through direct interactions between SHARP RRMs and Xist lncRNA[42–44] and between SHARP SPOC and Pol II CTD (Fig. 2c). In addition, intrinsically disordered regions (IDRs) in SHARP promote its accumulation on Xist lncRNA[45,46]. While the loss of SHARP RRMs and IDRs was previously shown to elicit its delocalization from Xist[12,45,46], here we showed that loss of the SPOC domain has a similar consequence (Fig. 8a). In SHARP ΔRRM cells, Xist lncRNA levels are reduced due to reduced RNA stability, which was not observed in SHARP ΔSPOC (Rodermund et al.[47], and our study). Loss of RRMs, IDRs, and SPOC impairs Xist lncRNA-mediated silencing during X-chromosome inactivation[12]. Although initially suggested to act via recruiting HDAC3, Xist-mediated silencing is abolished by SHARP deletion, but only attenuated upon HDAC3 deletion[48,49], suggesting that additional mechanisms are involved in SHARP-dependent repression of the X-chromosome. One of the earliest events upon Xist upregulation at the onset of X-chromosome inactivation is the exclusion of Pol II from the chromosome territory[50], but the underlying mechanism of Pol II exclusion remains unknown. Our finding that SHARP directly interacts with Pol II CTD pS5 opens up the intriguing possibility that SHARP might initiate gene silencing by directly acting on Pol II.

We furthermore identified FMR1 (FMRP) as a SPOC-dependent interactor of SHARP. An expansion of CGG-repeats within FMR1 leads to the reduction or abolishment of its expression and is the cause of fragile X syndrome, which can manifest as mild to severe intellectual disability and autism spectrum disorder[51]. On a molecular level, FMR1 is an RNA-binding protein that binds m$^6$A mRNA, regulates mRNA export, and acts as a translational repressor to modulate the amount of protein production[35,52]. We show that SHARP SPOC binds to an LSD motif in FMR1 in a phosphorylation-dependent manner. The physiological role of this interaction is yet to be resolved; however, SHARP and its SPOC domain have also been implicated in neurodevelopment[7,8] and SHARP and FMR1 might cooperate to ensure proper formation of neurons with SHARP controlling the transcriptional aspects and FMR1 regulating mRNA export and translation.

Interaction with the m$^6$A regulators may be a common theme for the Spen family of SPOC proteins. While SHARP interacts with the m$^6$A reader protein FMR1 in a SPOC-dependent manner, RBM15 SPOC is crucial for its interaction with the m$^6$A writer complex (Fig. 5d, e). RBM15 was previously shown to recruit the m$^6$A writer complex to RNA via WTAP and ZC3H13, but the mechanism remained unclear[16]. Our results suggest that RBM15 uses its SPOC domain to bind phosphorylated WTAP and recruit the m$^6$A machinery onto RNA bound by RBM15 RRMs (Fig. 9). Without RBM15 or its SPOC domain, cellular m$^6$A levels are globally reduced while transcripts are globally downregulated (Fig. 6g, h, k), suggesting that RBM15 positively regulates mRNA stability through the m$^6$A RNA modification. The m$^6$A RNA modification was previously shown to regulate mRNA decay and translation and may exert a positive or a negative effect on mRNA stability depending on m$^6$A reader proteins[53].

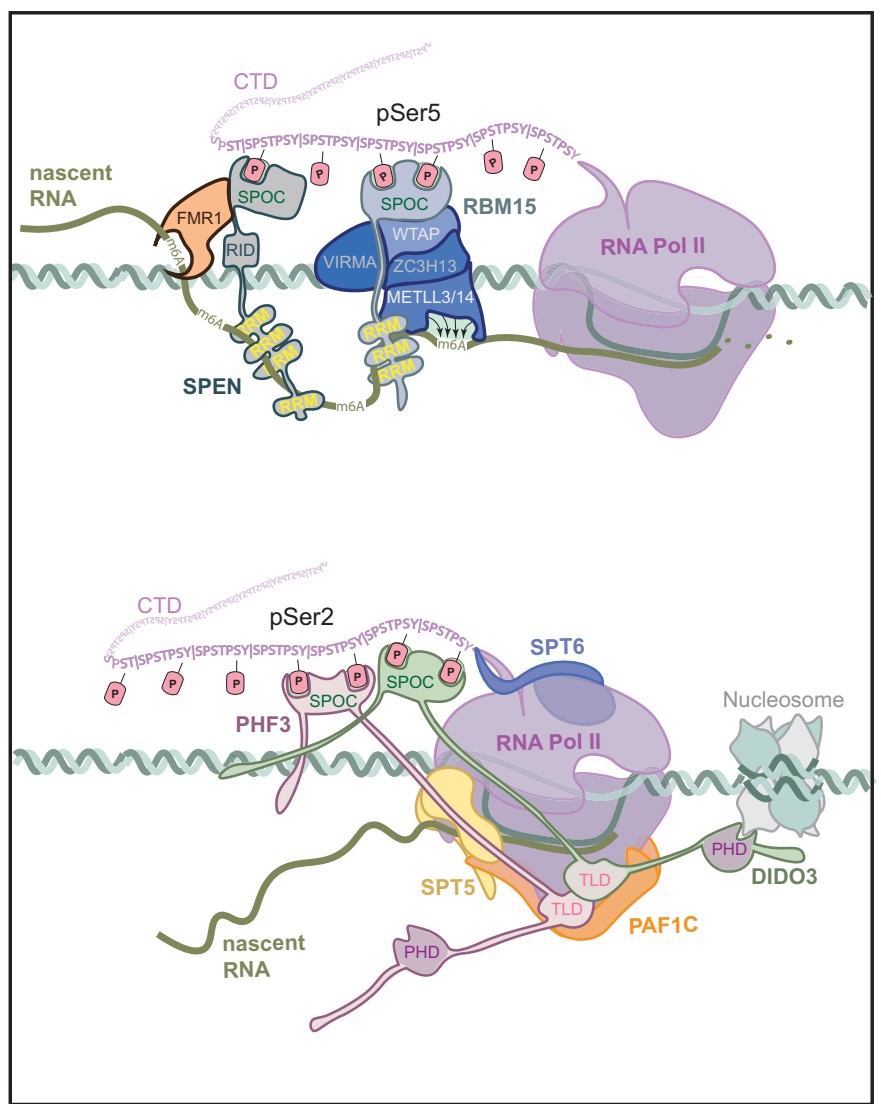

**Fig. 9 | SPOC bridges transcription with co-and post-transcriptional processes.** SHARP and RBM15 interact with RNA and writers and readers of m⁶A RNA mod-ification, while PHF3 and DIDO interact with regulators of transcription and co-transcriptional processing. SPOC protein interactions, in turn, regulate down-stream processes like translation and mRNA decay.

Analogous to PHF3- and DIDO-mediated co-regulation of transcription through SPOC-dependent binding to Pol II CTD, RBM15 and SHARP may cooperate in co- and post-transcriptional RNA processing through m⁶A RNA modification. Interestingly, SHARP interacts with RBM15 in a SPOC-dependent manner (Fig. 5a, d), suggesting that SHARP and RBM15 may work together to recruit the m⁶A writer complex onto target RNAs such as Xist lncRNA (Fig. 9). Xist RNA is modified with m⁶A, which contributes to Xist-mediated X-chromosome silencing[54]. Knock-out of RBM15 or m⁶A writer complex components shows reduced, but not completely abro-gated Xist m⁶A and a modest effect on X-chromosome silencing[49]. Although m⁶A levels were not globally altered in SHARP ΔSPOC (Fig. 6k), Xist m⁶A levels may be reduced, which remains to be fur-ther explored.

Taken together, we showed that SPOC domains from PHF3, DIDO, SHARP, and RBM15 have a different affinity and specificity towards CTD phosphomarks encoded in their distinct surface electrostatic potential patterns, suggesting that SPOC is a versatile Pol II CTD reader. Moreover, SPOC domains are versatile phosphoserine readers engaging with the Pol II transcription elongation complex, co-repressor complexes, m⁶A writer, and reader machinery. Multivalent interactions of SPOC domain proteins facilitate coupling between transcription and RNA metabolism to ensure appropriate gene expression.

## Methods

### Cloning

For mammalian expression, human DIDO3, SHARP, and RBM15 were amplified from HEK293T cDNA and cloned into CMV10 N3XFLAG (Sigma) by Gibson assembly (NEB). These constructs were used for cloning ΔSPOC truncations using Gibson assembly (NEB). CMV10 N3XFLAG-PHF3 and PHF3 ΔSPOC constructs were generated previously[5]. NLS-SPOC constructs were cloned into CMV10 N3XFLAG (Sigma) between NotI and XbaI. NLS sequence (RAPKKKRKVGG) was introduced to ensure nuclear localization of the isolated SPOC domains. For bacterial expression, human DIDO3 SPOC (1047–1205 aa), SHARP SPOC (3496–3664 aa), RBM15 SPOC (775–960 aa), and SPOCD1 SPOC (858–1025 aa) were cloned into pET M11 between NcoI and XhoI for N-terminal His6 fusion. His6-PHF3 SPOC was generated previously[5]. Arginine mutations were introduced by site-directed mutagenesis according to the Fas-tCloning protocol[55]. Repair templates for CRISPR/Cas9 knock-in

were cloned by Gibson assembly. Primer sequences are listed in Supplementary Table 4.

## Protein purification

SPOC domains were expressed in *E. coli* Rosetta2 (DE3) cells (Novagen). ON cultures were diluted 1:50 in 2×TY, grown at 37 °C until $OD_{600} = 0.8$, and induced with 0.5 mM IPTG for 3 h at 30 °C. $His_6$-tagged SPOC domains were purified by affinity chromatography using HisTrap HP column (Cytiva) or His-Pur Ni-NTA resin (Thermo Scientific) equilibrated in 25 mM Tris-Cl pH 7.4, 500 mM NaCl, 20 mM Imidazole. Elution with 25 mM Tris-Cl pH 7.4, 500 mM NaCl, and 500 mM Imidazole was followed by TEV cleavage of the $His_6$-tag and size exclusion chromatography using Sephacryl S-200 HR 16/600 or 26/600 (Cytiva) equilibrated in 25 mM Tris-Cl pH 7.4, 25 mM NaCl, 1 mM DTT.

## Peptide labeling

CTD peptides were N-terminally labeled with Atto-488, while NCoR and FMR1 peptides were purchased with an N-terminal FAM label. 0.5 mg of lyophilized CTD peptides were dissolved in 30 μL DMSO (Sigma-Aldrich). One molar equivalent of Atto-488 NHS ester (Attotec) and five equivalents of DIPEA (Sigma-Aldrich) was added to three equivalents of the peptide. Reactions were incubated ON at room temperature (RT) and protected from light. Labeled peptides were purified by reverse-phase HPLC over a C18 column (Agilent Technologies) using a gradient from 30 to 70% Methanol (Merck). Purified peptides were lyophilized and stored at −20 °C.

## Fluorescence anisotropy

Fluorescence anisotropy measurements were performed on a Perkin Elmer LS50 B fluorescence spectrometer operated by Perkin Elmar FL WinLab software (version 3.00). The excitation wavelength was set to 500 nm (excitation slit 10 nm), the emission wavelength to 518 nm (emission slit 6 nm), and the integration time to 1 s. Measurements were conducted at 20 °C in 25 mM Tris-Cl pH 7.4, 25 mM NaCl, and 1 mM DTT. The grating factor was determined using unconjugated Atto-488 dye. Totally, 90 nM peptides were titrated with increasing concentrations of SPOC domains in a final volume of 110 μL. Each individual data point is an average of 20 measurements. For each binding curve, data points were acquired from three independent SPOC dilution series. Data are presented as mean anisotropy ± standard deviation from the three independent replicates. Data were plotted and fitted with QtiPlot 1.0.0-rc13 software (version 5.9.8.).

## X-ray crystallography

Crystallization was performed at 22 °C or 4 °C using a sitting-drop vapor diffusion technique and micro-dispensing liquid handling robot Mosquito (TTP labtech). The best diffracting crystals of SHARP SPOC:1×S5P CTD were grown at 22 °C in conditions E9 from ShotGun HT screen (SG1 HT96 Molecular Dimensions, Suffolk, UK) containing 0.1 M potassium thiocyanate, 30% w/v PEG 2000 MME at protein concentration 10 mg/ml. The best diffracting crystals of RBM15 SPOC were grown at 4 °C in conditions F12 from PACT screen (PACT premier HT96 Molecular Dimensions, Suffolk, UK) containing 0.2 M Sodium malonate dibasic monohydrate, 0.1 M Bis−Tris propane pH 6.5, 20% w/v PEG 3350 at protein concentration 6 mg/mL. The crystals were flash cooled in liquid nitrogen prior to data collection. The data set of SHARP SPOC:1×S5P CTD was collected at the MASSIF beamline ID30a1 (ESRF, Grenoble) at 100 K using a wavelength of 0.966 Å. The data set of RBM15 was collected at the I04 (DLS, Didcot Oxfordshire) 100 K using a wavelength of 0.9795 Å. The data frames were processed using the XDS package[56] and converted to mtz format with the program AIMLESS[57] with the help of autoprocessing pipelines at the synchrotrons. The structures were determined using the molecular replacement program PHASER with atomic coordinates of SPOC SHARP (PDB

code 1OW1) as a search model in the case of SHARP SPOC:1×S5P structure and our SHARP SPOC structure as a search model for RBM15. The structures were then refined with Phenix Refine v 1.20.1−4487[58] and rebuilt using Coot v 0.9.6[59]. Structural depictions were generated using UCSF Chimera (version 1.14)[60]. The atomic coordinates were deposited in the Protein Data Bank under accession codes: 7Z27 for RBM15 SPOC and 7Z1K for SHARP SPOC:1×S5P CTD.

## Modeling of SPOC complexes with peptides

3D configurations of complexes of SHARP SPOC with all peptides in phosphorylated and unphosphorylated forms (except the CTD fragment) were generated using PyMOL (version 2.5.2) starting from the SHARP-SMRT structure (2RT5)[3] as a template. The X-ray structures refined in the present study were used as templates for complexes involving phosphorylated CTD 8-mer repeat (SYSPTSPS) and SHARP and RBM15 SPOC domains, whereby the CTD peptide was extended to eight residues. The structure of RBM15 SPOC predicted by AlphaFold (see https://alphafold.ebi.ac.uk/entry/Q96T37) was also used for this purpose. All complexes were further refined using HADDOCK 2.2 webserver[28,29]. The peptide length was fixed to eight residues in all cases. The HADDOCK score, which was calculated after model refinement for different complexes, was used as a proxy for stability (Fig. 3f).

## Size exclusion chromatography and multiangle laser light scattering (SEC-MALLS)

SEC-MALLS was performed with a Superdex 75 Increase 10/300 GL column (Cytiva) on an Agilent Technologies 1260 Infinity HPLC equipped with a miniDawn Treos detector (Wyatt Technology) using a laser emitting at 690 nm. Experiments were performed at RT with a flow rate of 0.5 mL/min in 25 mM Tris pH 7.4, 25 mM NaCl, and 1 mM DTT. 80 μL protein samples at a concentration of 2.5–6 mg/mL were injected. The HPLC was operated with OpenLAB CDS software (Agilent Technologies, Rev C.01.07 SR3 [465]), MALLS data acquisition and analysis were performed with ASTRA software (Wyatt Technology, version 7.3.2.19).

## Cell culture

HEK293T cells were grown in Dulbecco's Modified Eagle's Medium (DMEM 4.5 g/L glucose) (Sigma) supplemented with 10% fetal bovine serum (Sigma), 1% L-glutamine (Sigma), 1% penicillin-streptomycin (Sigma) under 5% $CO_2$ at 37 °C. Drosophila S2 cells were grown in Schneider's Drosophila Medium (Gibco) supplemented with 10% fetal bovine serum (Sigma) at 28 °C. To generate endogenously tagged SHARP-GFP and SHARP ΔSPOC-GFP cell lines by CRISPR/Cas9, gRNAs targeting the 3'end of *SHARP* (SHARP 3'gRNA: 5'- CCACTCAGTGGCT CACACGG-3') and the region upstream of SPOC (SHARP SPOC gRNA: 5'-GAACCATATCCACGGGTCTC) were designed and cloned into pX330 plasmid encoding Cas9 nuclease[61]. HEK293T cells were transfected with 2 μg pX330-SHARP 3'gRNA and 4 μg plasmid repair template comprising EGFP-P2A-puromycin and flanking ≈1 kb homology arms for SHARP-GFP or 2 μg pX330-SHARP 3'gRNA + 2 μg pX330-SHARP SPOC gRNA to excise the region encoding SPOC and 4 μg plasmid repair template for SHARP ΔSPOC-GFP. Two days after transfection, 0.5 μg/μL puromycin was added to the culture medium for 1 week. To allow for recovery, surviving cells were grown for several days without puromycin. GFP-positive cells were sorted into 96-well plates by FACS. Colonies originating from single cells were expanded, genomic DNA was extracted, and positive clones were identified by PCR and sequencing. To generate an endogenously tagged PHF3 ΔSPOC-GFP cell line, HEK293T PHF3 ΔSPOC cells were transfected with 2 μg each of pX335 plasmids encoding Cas9 nickase and gRNAs targeting the *PHF3* 3'end (5'-CAGTGTGGTCCCTATCTTTG-3' and 5'-TAAAATTTGCAGGCTGCTTC-3) and 4 μg plasmid repair template containing EGFP-P2A-puromycin and flanking 1.5 kb homology arms. The PHF3 ΔSPOC cell line and plasmids had been generated for a previous study[5]. Selection and identification of positive clones were

performed as described above. For DIDO KO and RBM15 KO, gRNAs were cloned into pX458 plasmid encoding Cas9 nuclease and EGFP (DIDO: 5′-TGGGCATTTCTGAGGCTCGA-3′ targeting exon 7; RBM15: 5′-ACTCGACTTCCCGCGGTGAG-3′ targeting exon 1). 70% confluent cells were transfected with Cas9-EGFP gRNA plasmid (15.6 µg/10 cm dish), followed by FACS sorting of GFP-positive cells after 24–72 h. After 1 week, GFP-negative cells were FACS-sorted 1 cell/well in 96-well plates. Genomic DNA was isolated using QuickExtract (Lucigen) and Cas9 target region was amplified by PCR and sequenced. To generate DIDO and RBM15 SPOC-deleted cell lines, two gRNAs targeting the flanking region of the SPOC domain (DIDO: 5′-GGGTCGTGTCTCCCTCTGGA-3′ and 5′-CCAAGAATTATATTCGGACG-3′; RBM15: 5′-CCCATCCTGTTTCTGGGACG-3′ and 5′-TGGCGCTGACCCTGTTATAG-3′) were cloned into the pX458 plasmid. Plasmid-borne repair template consisted of either Hygromycin, Puromycin, or Blasticin resistance genes placed into an intron in antisense orientation flanked by 999 bp homology arms corresponding to the targeted genomic region. 1 million cells were electroporated with 2 µg of each Cas9 gRNA plasmid and 10 µg of the repair template. 72 h after electroporation, 0.5 µg/mL puromycin was added to the culture medium. After 1 week, selection media was replaced by a full medium without selective antibiotics and cells were allowed to recover for 3 days. Cells were subsequently FACS-sorted 1 cell/well in 96-well plates. After two weeks, surviving clones were expanded in culture, genomic DNA was isolated, and the target region was PCR-amplified and sequenced. FACS sorting during cell line generation was performed on a FACS Melody instrument (BD) operated with BD FACSChorus software (version 1.1.20.0). The gating strategy is illustrated in Supplementary Fig. 15.

## Immunoprecipitation

For anti-FLAG immunoprecipitation, HEK293T cells were transfected with the following FLAG constructs: 3×FLAG-NLS-PHF3 SPOC, 3×FLAG-NLS-DIDO3 SPOC, 3×FLAG-NLS-SHARP SPOC, 3×FLAG-NLS-RBM15-SPOC, 3×FLAG-PHF3, 3×FLAG-PHF3 ΔSPOC, 3×FLAG-SHARP, 3×FLAG-SHARP ΔSPOC, 3×FLAG-DIDO3, 3×FLAG-DIDO3 ΔSPOC, 3×FLAG-RBM15, 3×FLAG-RBM15 ΔSPOC. Cells were seeded on a 10 cm dish one day prior to transfection, transfected with 8 µg of the respective plasmid at 60–70% confluency, and harvested 48 h after transfection. For anti-GFP immunoprecipitation, one 10 cm dish of HEK293T cell lines expressing GFP-tagged SHARP WT or ΔSPOC was used per immunoprecipitation. Pellets were lysed in 1 mL lysis buffer (50 mM Tris-Cl pH 8, 150 mM NaCl, 0.1% Triton, 1× protease inhibitors, 2 mM Na₃VO₄, 1 mM PMSF, 2 mM NaF, 50 units/ml benzonase and 1 mM DTT) for 30 min on a rotating wheel at 4 °C. Protein concentrations were measured using the Bradford protein assay using NanoDrop 2000c. The volume used for IP was adapted to the lysate with the lowest measured protein concentration to ensure that the same amount of protein lysate was used for immunoprecipitation. Lysates were incubated for 2 h on a rotating wheel at 4 °C with anti-FLAG M2 beads (Sigma). Beads were washed once with lysis buffer (without benzonase) and 4 times in TBS. For mass spectrometry analysis, the samples were further processed as described in Mass spectrometry sample preparation. For western blot analysis of anti-FLAG immunoprecipitation, the beads were eluted with 30 µL of 3×FLAG peptide (150 ng/µL) in TBS for 30 min on a rotating wheel at 4 °C. Ten microlitres of the 4×SDS loading buffer was added to the eluate and half of the total volume (20 µL) was analyzed by SDS-PAGE followed by western blotting. Western Blots were imaged on a ChemiDoc MP Imaging system (Bio-Rad) operated by Bio-Rad Image Lab Touch Software (version 2.3.0.07) and analyzed using Bio-Rad Image Lab Software (version 5.2.1). Antibodies used for Western Blotting are listed in Supplementary Table 3.

## Mass spectrometry sample preparation for protein analysis

Magnetic anti-FLAG beads were transferred to fresh tubes and resuspended in 50 µL of 50 mM ammonium bicarbonate (ABC). The proteins were digested with 200 ng Lys-C (WAKO) at 37 °C shaking at 1200 rpm for 3 h. The supernatant was transferred to a fresh tube (L-fraction). Still-bound proteins were eluted from the beads by adding 20 µL of 100 mM glycine pH 2. After gentle vortexing and incubation at RT for 2–5 min, the supernatant was transferred to a fresh tube (G-fraction). The elution step was performed three times in total, the eluates were combined and the pH was made alkaline using 1 M Tris-HCl pH 8. L- and G-fractions were subsequently processed in parallel. Disulfide bonds were reduced with 10 mM dithiothreitol for 45 min at 56 °C. Alkylation was performed with 20 mM iodoacetamide at RT for 45 min in the dark. The remaining iodoacetamide was quenched by adding 5 mM DTT and the proteins were digested with 200 ng trypsin (Trypsin Gold, Promega) at 37 °C ON. The digest was stopped by the addition of trifluoroacetic acid (TFA) to a final concentration of 0.5 %, and the peptides were desalted using C18 Stagetips. After quality checks of the digests by liquid chromatography, L- and G-fractions from the respective sample were combined. Beads with cross-linked anti-GFP nanobody were transferred to fresh tubes and resuspended in 30 µL of 2 M urea in 50 mM ammonium bicarbonate (ABC). Disulfide bonds were reduced with 10 mM dithiothreitol for 30 min at RT before adding 25 mM iodoacetamide and incubating for another 30 min at RT in the dark. The remaining iodoacetamide was quenched by adding 5 mM DTT and the proteins were digested with 150 ng trypsin (Trypsin Gold, Promega) for 90 min at RT in the dark. The supernatant was transferred to a fresh tube. The beads were rinsed with 30 µL of 2 M urea in 50 mM ABC. The combined supernatants were diluted to 1 M urea with 50 mM ABC, additional 150 ng trypsin (Trypsin Gold, Promega) was added and incubated at 37 °C ON. The digest was stopped by the addition of 10% trifluoroacetic acid (TFA) to a final concentration of 0.5%, and the peptides were desalted using C18 Stagetips[62].

## Liquid chromatography-mass spectrometry (LC–MS) of peptides

LC–MS analysis included 15 FLAG-IP samples and nine GFP-IP samples. An empty vector was used as a control for FLAG-IP, and WT cells were used as a control for GFP-IP. Experiments were performed in three biological replicates. Peptides were separated on an Ultimate 3000 RSLC nano-HPLC system using a pre-column for sample loading (Acclaim PepMap C18, 2 cm × 0.1 mm, 5 µm), and a C18 analytical column (Acclaim PepMap C18, 50 cm × 0.75 mm, 2 µm, all HPLC parts Thermo Fisher Scientific), applying a linear gradient from 2 to 35% solvent B (80% acetonitrile, 0.1% formic acid; solvent A 0.1% formic acid) at a flow rate of 230 nL/min over 120 min. The GFP-IPs were analyzed on a Q Exactive HF-X Orbitrap, and the FLAG-IPs on an Orbitrap Fusion Lumos mass spectrometer coupled to the HPLC via the EASY-Spray ion source (all Thermo Fisher Scientific) equipped with coated emitter tips (New Objective). The mass spectrometers were operated in data-dependent acquisition mode (DDA). On the Q Exactive HF-X survey, scans were obtained in a mass range of 375–1500 m/z with lock mass off, at a resolution of 1,20,000 at 200 m/z and an AGC target value of 3E6. The 8 most intense ions were selected with an isolation width of 1.6 m/z, for max. 250 ms at a target value of 1E5, and then fragmented in the HCD cell at 28 % normalized collision energy. Spectra were recorded at a resolution of 30,000. Peptides with a charge of +1 or >+6 were excluded from fragmentation, the peptide match feature was set to preferred, the exclude isotope feature was enabled, and selected precursors were dynamically excluded from repeated sampling for 30 s. On the Orbitrap Lumos Fusion, the survey scans were obtained in a mass range of 375–1500 m/z, at a resolution of 1,20,000 at 200 m/z, with an AGC target value of 4E5. In a cycle time window of 2.5 s, the most intense precursors were selected with an isolation width of 1.0 m/z, aiming at an AGC target value of 2E5 within 150 ms, and fragmented in the HCD cell with a collision energy of 30%. The spectra were recorded in the orbitrap at a resolution of 30,000.

Peptides with a charge of +2 to +6 were included for fragmentation, the MIPS mode was set to "peptide" and the exclude isotope feature was enabled. Selected precursors were dynamically excluded from repeated sampling for 45 s.

## Peptide mass spectrometry data analysis

Raw data were processed using the MaxQuant software package[63] (version 1.6.16.0 respect. 1.6.14.0) searching against the Uniprot human reference proteome (January 2020, www.uniprot.org) as well as a database of most common contaminants. The search was performed with full trypsin specificity and a maximum of two missed cleavages. Carbamidomethylation of cysteine residues was set as fixed, oxidation of methionine, and acetylation of protein N-termini as variable modifications. For the FLAG-IP samples, acetylation of lysines and phosphorylation of serine, threonine, and tyrosine were additionally defined as variable modifications. For label-free quantification, the "match between runs" feature and the LFQ function were activated−all other parameters were left at default. Results were filtered at a false discovery rate of 1% at protein and peptide spectrum match levels. MaxQuant output tables were further processed in R[64] (version 4.0.2). Reverse database identifications, contaminant proteins, protein groups identified only by a modified peptide, protein groups with less than two quantitative values in one experimental group, and protein groups with less than 2 razor peptides were removed for further analysis. Missing values were replaced by randomly drawing data points from a normal distribution model on the whole dataset (data mean shifted by −1.8 standard deviations, the width of the distribution of 0.3 standard deviations). Differences between groups were statistically evaluated using the LIMMA package[65] at 5% FDR (Benjamini−Hochberg). Mass spectrometry data have been deposited to the jPOST repository[66] under the accession numbers JPST001505 (anti-FLAG IP) and JPST001502 (anti-GFP IP).

## RNA isolation and RNA-seq library preparation

RNA-seq samples were prepared in three biological replicates. Cells were harvested and counted, 20% Drosophila S2 cells were added as a spike-in control. Cell pellets were resuspended in 1 mL TRI reagent (Sigma). 200 μL Chloroform (Applichem) was added, and samples were mixed and centrifuged at 4 °C, max. speed for 15 min. The upper phase was transferred to a fresh tube and subjected to isopropanol precipitation. 20 μg of RNA were treated with 40 U DNase I (Roche) at 37 °C for 30 min and purified by phenol−chloroform extraction and Ethanol precipitation. rRNA was depleted. DIDO and SHARP RNA-seq libraries were prepared with NEBNext rRNA depletion kit v2 (Human/Mouse/Rat) and NEBNext Ultra II Directional RNA Library Prep Kit for Illumina (New England Biolabs) according to the manufacturer's instructions. RBM15 RNA-seq libraries were prepared using Lexogen RiboCop rRNA Depletion kit HMR V2 and Lexogen CORALL Total RNA-seq Library Prep kit according to the manufacturer's instructions. Totally, 600 ng total RNA was used as an input for rRNA depletion. Sequencing was performed on an Illumina NovaSeq 6000 instrument in readmode SR100 or PE150 by the Next Generation Sequencing facility at Vienna BioCenter Core Facilities (VBCF).

## TT$_{chem}$-seq

TT$_{chem}$-seq was performed as described[37] with some adjustments. Cells were counted and 5 million cells were seeded in 10 cm dishes the day before the experiment. Cells were labeled at 70% confluency by the addition of 1 mM 4sU (Glentham Life Sciences) for 15 min. Labeling was stopped and cells were lysed by the addition of 1 mL TRI reagent (Sigma) directly to the culture dish. Cells from additional dishes seeded in the same way were counted to estimate cell number. 2.4 ng in vitro transcribed synthetic spike-in RNA per million cells were added directly to the TRI reagent lysate. Totally, 200 μL Chloroform

(Applichem) was added to the lysate, samples were mixed and centrifuged at 12,000$g$, 4 °C for 15 min. The aqueous phase was mixed with an equal volume of Chloroform:Isoamyl alcohol 24:1 (Applichem) and centrifuged at 12,000$g$, 4 °C for 5 min. RNA in the aqueous phase was precipitated by Isopropanol precipitation and reconstituted in 100 μL H$_2$O. 1 μg 4TU-labeled yeast RNA was added to 100 μg RNA as a spike-in control in a total volume of 100 μL. RNA was fragmented by the addition of 20 μL of 1 M NaOH for 20 min on ice, the reactions were neutralized and fragmented RNA was purified using Micro Bio-Spin P-30 columns (Biorad). 4sU-labeled RNA was biotinylated by the addition of 5 μg MTSEA biotin-XX linker for 30 min at RT protected from light. After biotinylation, RNA was purified by Phenol−Chloroform extraction and isopropanol precipitation. Biotinylated fragments were enriched using the μMACS Streptavidin kit (Miltenyi Biotec). Sequencing libraries were prepared using the NEBNext Ultra II Directional RNA Library Prep Kit for Illumina (New England Biolabs) according to the manufacturer's instructions. Sequencing was performed on an Illumina NovaSeq 6000 instrument in readmode PE150 by the Next Generation Sequencing facility at Vienna BioCenter Core Facilities (VBCF).

## Preparation of spike-ins for TT$_{chem}$-seq

In vitro transcribed synthetic spike-in RNA was prepared as described[67]. One microlitre ERCC RNA Spike-in Mix (Invitrogen) were reverse transcribed using Protoscript II reverse transcriptase (New England Biolabs). cDNA was diluted 1:4 in H$_2$O and used as a template for PCR amplification of ERCC spike-ins 00043, 00170, 00136, 00145, 00092, and 00002. PCR products were purified and verified by agarose gel electrophoresis and Sanger sequencing. 0.5 μg PCR product was used as an input for in vitro transcription using the MEGAscript T7 transcription kit (Invitrogen) according to the manufacturer's instructions. For spike-ins 00043, 00136, and 00092, 10% UTP was substituted with 4-Thio-UTP. RNA spike-ins were purified using AMPure XP beads (Beckman Coulter), quantified using Qubit HS RNA Assay kit (Invitrogen), and mixed at a final concentration of 1 ng/μL of each spike-in RNA. The final spike-in mix was quantified again using the Qubit HS RNA Assay kit (Invitrogen) and stored at −80 °C in single-use aliquots. For the preparation of 4TU-labeled yeast RNA, an ON culture of *S. cerevisiae* BY4741 was diluted to OD$_{600}$ = 0.1 in 50 mL YPD + 2% (w/v) glucose at grown at 30 °C, 180 rpm until OD$_{600}$ = 0.8. 4TU was added to a final concentration of 5 mM, and cells were labeled for 5 min. Labeled cells were pelleted, resuspended in 250 μL 0.8 M sorbitol, 0.1 M EDTA, 0.1% β-mercaptoethanol, 200 U/mL lyticase and incubated at 30 °C for 30 min. The lysate was mixed with 750 μL TRI Reagent LS (Sigma), 200 μL Chloroform (Applichem) was added, and samples were mixed and centrifuged at 4 °C, max. speed for 15 min. The aqueous phase was subjected to isopropanol precipitation. RNA was treated with DNase I (Roche) at 37 °C for 30 min and purified by phenol−chloroform extraction and Ethanol precipitation.

## RNA-seq data analysis

RNA-seq data from HEK293 cells from different genetic backgrounds were processed using the PiGx-RNA-seq[68] pipeline. In short, the data was quantified using the GRCh38/hg38, and the dm6 versions of the human and drosophila spike-in transcriptome (downloaded from the ENSEMBL database[69]) using SALMON[70] with default parameters. For visualization purposes, the data was mapped to the GRCh38/hg38, and dm6 versions of the human, and drosophila genomes using STAR, with the following parameters: --limitOutSJcollapsed 20000000 --limitIObufferSize=1500000000 --outFilterMultimapNmax 10 --seed-PerWindowNmax 5. The quantified data was processed using tximport[71], and the differential expression analysis was done using DESeq2[72]. Genes with less than 5 reads in all biological replicates of one condition were filtered out before the differential analysis. The data

was normalized by taking the ratio of reads mapping to the human and the drosophila transcriptome. Genes were defined as differentially expressed if they had a minimum absolute log2 fold change of 1, and a BH adjusted p value less than 0.05. RNA-seq data were deposited under the accession number E-MTAB-12358.

## TT$_{chem}$-seq data analysis

TT$_{chem}$-seq data from HEK293T cells from different genetic backgrounds were processed using the PiGx-RNA-seq[68] pipeline. In short, the data was quantified using the GRCh38/hg38, and the dm6 versions of the human, and labeled and unlabeled spike-in sequences (downloaded from the ENSEMBL database[69]) using SALMON[70] with default parameters. The data were mapped to the GRCh38/hg38 version of the human genome using STAR, with the following parameters: --limitOutSJcollapsed 20000000 --limitIObufferSize = 1500000000. The STAR genome index was created using the following parameter: --genomeSuffixLengthMax 300. To calculate the changes in transcriptional initiation and the stalling index, the data was quantified over the TSS and gene bodies. The TSS region was defined as ±250 base pairs around the TSS of the gene promoters, as defined in the ENSEMBL gene annotation for hg38. Gene bodies were defined as a region +251 to the end of the gene. Genes shorter than 500 base pairs were removed from the analysis. Fragments overlapping both the TSS and the gene body were not counted. The differential initiation and stalling index were calculated using DESeq2[72]. Genes with less than 5 reads in all biological replicates of one condition were filtered out before the differential analysis. The data was normalized by taking the ratio of reads mapping to the human to the median of labeled spike in. Genes were defined as differentially expressed if they had a minimum absolute log2 fold change of 1, and a BH adjusted p value less than 0.05. TT$_{chem}$-seq data was deposited under the accession number E-MTAB-12359.

## RNA isolation and mRNA enrichment for m$^6$A mass spectrometry analysis

mRNA samples were prepared in three biological replicates. Cells were grown in 15 cm dishes to 70–80% confluency. Cell pellets were resuspended in 1.5 mL TRI reagent (Sigma). 300 μL Chloroform (Applichem) was added, and samples were mixed and centrifuged at 4 °C, max. speed for 15 min. The upper phase was transferred to a fresh tube and subjected to isopropanol precipitation. Totally, 120 μg of RNA were treated with 100 U DNase I (Roche) at 37 °C for 30 min and purified by phenol-chloroform extraction and Ethanol precipitation. 100 μg total RNA was used as an input for mRNA enrichment. Polyadenylated RNA was isolated using Dynabeads mRNA Purification Kit (Invitrogen) according to the manufacturer's instructions. rRNA was depleted using NEBNext rRNA depletion kit v2 (Human/Mouse/Rat) (New England Biolabs) according to the manufacturer's instructions, followed by another round of mRNA enrichment using Dynabeads mRNA Purification Kit (Invitrogen). The absence of rRNA in the final mRNA preparations was confirmed by capillary electrophoresis on a Fragment Analyzer (Supplementary Fig. 14).

## RNA hydrolysis to nucleosides

Totally, 100–200 ng mRNA were hydrolyzed to nucleosides in 5 mM Tris pH 8, 1 mM MgCl$_2$, 1 U benzonase, 0.1 U phosphodiesterase I, 1 U calf intestinal phosphatase, 1 μg pentostatin, 5 μg tetrahydrouridine and 10 μM butylated hydroxytoluene in a total volume of 35 μl. Samples were incubated at 37 °C for 2 h. After hydrolysis, 10 μL of 5 mM NH4OAc pH 5.3 were added to each sample for acidification.

## LC−MS of hydrolyzed nucleosides

LC−MS experiments were performed on an Agilent 1290 Infinity II equipped with a Phenomenex Synergi 2.5 μm Fusion-RP 100 Å (100 × 2 mm) coupled to an Agilent 6470 Triple Quad equipped with electron spray ionization. Of each sample, 18 μL were injected without

prior filtering. Chromatographic separation was carried out at 35 °C with a flow rate of 0.35 ml/min using a linear gradient of two solvents: 5 mM ammonium acetate pH 5.3 as solvent A and acetonitrile as solvent B (gradient: 0–1 min hold at 0% B, 1–5 min increase to 10% B, 5–7 min increase to 40% B, 7–8 min hold at 40% B, 8–8.5 min decrease to 0% B, 8.5–11 min hold at 0% B). Optimized MS parameters for each compound can be found in Supplementary Data 6. Quantification was done using calibration curves of synthetic standards and stable isotope labeled internal standards (20 ng in 1 μL was automatically added by the instrument per sample) for each nucleoside[73] using Agilent's MassHunter software (version 9.0.647.0). To obtain the calibration curves, a solution containing synthetic standards of all nucleosides was serially diluted by factor 1:2 (twelve calibration levels). The highest injected amounts were 100 pmol (C, U, G, A), 20 pmol (Ψ), or 5 pmol (all other nucleosides).

## Immunofluorescence

Glass coverslips (thickness #1.5, diameter 12 mm, sterilized by baking overnight at 180 °C) were pretreated with 10 μg/mL fibronectin (Sigma, F1141) for 3 h at RT to ensure enhanced cell adhesion. Cells were seeded onto coverslips and grown to a confluency of about 80% before fixation. Cells were washed once with PBS, fixed with 4% paraformaldehyde (Sigma) for 10 min, and washed again three times before permeabilization in 0.5% Triton X-100 (Sigma) for 6 min. After washing three times with PBS, cells were blocked in blocking buffer (0.1% Tween + 3% bovine serum albumin (BSA) in PBS, Sigma P1379 & A4503) for at least 20 min at RT. Incubation with primary antibodies mouse anti-FLAG (1:500, M2 Sigma F1804), rabbit anti-GFP (1:1000, Abcam ab290), and Rabbit-anti RBM15 (1:200, Bethyl A300-821A) was done for 2 h at RT, washed three times, followed by secondary antibody 1:500 anti-mouse Alexa Fluor 568 and anti-rabbit Alexa Fluor 568, respectively for 1 h at RT in the dark. All coverslips were washed two times, stained with DAPI (1:10,000) for 5 min at RT, washed 1× with PBS and 1× with ddH$_2$O, and mounted onto slides with Prolong Diamond (Invitrogen, P36961). Immunofluorescence images were acquired using an inverse point scanning confocal Zeiss LSM980 Microscope equipped with a Zeiss Plan-Apochromat 63x/1.4 Oil DIC M27 (WD 0.19 mm) running with Zeiss ZEN Blue 3.3 software (version 3.3.89.0008). Sequential acquisitions of 2 channels were performed with a 405 nm laser diode (30 mW) and a 561 nm DPSS laser (25 mW) set to 1.5–1.8% excitation power for AF568 channel and 0.9% for DAPI, together with detector gain set to 750–850 V. Secondary beam splitters (SBS LP 570 for AF568 and SBS SP 550) were used to constrain emission wavelengths and pixel dwell time set to 1.99 μs. Detection was done with an Airyscan 2 detector (32 GaAsP elements). The field of views was acquired as 8-bit, 2114 × 2114 px images in unidirectional mode with a pixel size of 40 nm, detector offset of 0, and detector digital gain of 1.0. DAPI staining was used to identify the nuclei, and laser power and detector gain were balanced for each channel to enhance the signal intensity and reduce background noise, all being optimized according to Nyquist sampling. The Airyscan images were processed for super-resolution with Zen Blue 3.3 (version 3.3.89.0008) Auto Airyscan filter and furthermore thresholded using Fiji/ImageJ software (version 2.1.0/1.52c) with Costes-related automatic thresholds for each channel in each experiment for better digital and analog display.

## Chromatin immunoprecipitation

Cells were resuspended in 50 mL PBS/10$^8$ cells and fixed by adding 1% formaldehyde for 10 min. Formaldehyde was quenched by the addition of 0.6 M glycine pH 3 for 15 min. Cells were washed twice in cold PBS. Nuclei were isolated by resuspending the cells in 5 mL cold lysis buffer 1 (50 mM HEPES/KOH pH 7.5, 140 mM NaCl, 1 mM EDTA, 10% glycerol, 0.5% Nonidet P-40, 0.25% Triton X-100, 1× Complete protease inhibitors (Roche)) per 10$^8$ cells and rotating at 4 °C for 10 min. After centrifugation, nuclei were resuspended in

5 mL/$10^8$ cells cold lysis buffer 2 (10 mM Tris-Cl pH 8, 200 mM NaCl, 1 mM EDTA, 0.5 mM EGTA, 1× Complete protease inhibitors (Roche)) and rotated for 10 min at RT. The pellet, after centrifugation, was resuspended in 3 mL lysis buffer 3 (10 mM Tris-Cl pH 8, 100 mM NaCl, 1 mM EDTA, 0.5 mM EGTA, 0.1% Na-deoxycholate, 0.5% N-lauroylsarcosine, 1x Complete protease inhibitors (Roche)) and chromatin was sheared to an average size of 200–600 bp by sonication for 20 cycles 30 s on/30 s off using Bioruptor Pico (Diagenode). 1% Triton X-100 was added after sonication. 5–10% of chromatin was kept as input control for qPCR. For ChIP-seq, 1.5% spike-in chromatin from a mouse cell line expressing endogenously tagged PHF3-GFP was added. Anti-GFP antiserum (Abcam ab290) was added to sheared chromatin and rotated ON at 4 °C. Protein A Dynabeads (Invitrogen) were washed in cold block solution (0.5% BSA in PBS) three times, mixed with antibody-bound chromatin and rotated for 4–6 h at 4 °C. Beads were washed in RIPA washing buffer (50 mM HEPES/KOH pH 7.5, 500 mM LiCl, 1 mM EDTA, 1% NP-40, 0.7% Na-deoxycholate) five times and in 50 mM NaCl in TE once. Immunoprecipitated protein-DNA complexes were eluted in 200 μL elution buffer (50 mM Tris-Cl pH 8, 10 mM EDTA, 1% SDS) for 15 min at 65 °C. Eluates were incubated ON at 65 °C to reverse crosslinks and treated with 0.2 mg/mL RNase A for 2 h at 37 °C and 0.2 mg/mL proteinase K and 5.25 mM $CaCl_2$ for 30 min at 55 °C. DNA was purified by phenol-chloroform extraction followed by ethanol precipitation and resuspended in 50 μL nuclease-free water.

### ChIP-seq library preparation and sequencing
ChIP-seq libraries were prepared using NEBNext Ultra II DNA library prep kit for Illumina (New England Biolabs) and NEBNext Multiplex Oligos for Illumina (New England Biolabs) according to the manufacturer's instructions. Totally, 5–10 ng ChIP-DNA was used as input for library prep. Sequencing was performed on an Illumina NextSeq 550 instrument in readmode SR75 by the Next Generation Sequencing facility at Vienna BioCenter Core Facilities (VBCF).

### qPCR
qPCR analysis of input and ChIP-DNA was performed on a BioRad CFX Touch cycler operated by CFX Maestro software (version 2.2) using Takyon No Rox SYBR MasterMix dTTP Blue (Eurogentec). qPCR primer sequences are indicated in Supplementary Table 4. Input Cq values were adjusted to 100%, % input values were calculated as follows:

$$\% \text{ input} = 100*2^{\text{adjusted}Cq_{\text{input}} - Cq_{\text{ChIP}}}$$

qPCR data were analyzed and plotted using Microsoft Excel 365 (version 16.58) and GraphPad Prism 9 (version 9.2.0). Data are presented as mean ± standard deviation of four replicates. $p$-Value was calculated using a one-tailed, two-sample equal variance $t$-test. $p$-Values smaller than 0.05 were considered statistically significant, the level of significance is indicated with asterisks ('*' for $p < 0.05$; '**' for $p < 0.01$; '***' for $p < 0.001$; '****' for $p < 0.0001$).

### ChIP-seq analysis
ChIP seq data were processed using the PiGX−ChIPSeq pipeline[68]. In short, the data was mapped to the hg38 version of the human genome using Bowtie2[74], with the $k = 1$ parameter. The data were quantified over TSS and gene bodies as described for the TTseq data. Genes shorter than 2 kb were filtered out from the analysis. Genes with less than 5 reads in the TSS region or the gene body, in all biological replicates of one condition, were filtered out of the analysis. The data were normalized using DESeq2-derived size factors. Region width was used as an additional normalization factor. Differentially bound regions were defined using DESeq2[72], with the default parameters. ChIP-seq data were deposited under the accession number E-MTAB-11506.

### Reporting summary
Further information on research design is available in the Nature Portfolio Reporting Summary linked to this article.

## Data availability
The data that support this study are available from the corresponding author upon reasonable request. The atomic coordinates have been deposited in the Protein Data Bank under accession codes: 7Z27 [https://doi.org/10.2210/pdb7Z27/pdb] for RBM15 SPOC and 7Z1K [https://doi.org/10.2210/pdb7Z1K/pdb] for SHARP SPOC:1xS5P CTD. The sequencing data generated in this study have been deposited in ArrayExpress under accession codes: E-MTAB-12358 (RNA-seq), E-MTAB-12359 (TT$_{chem}$-seq), E-MTAB-11506 (ChIP-seq). The processed sequencing data are provided in Supplementary Data 3–5. Mass spectrometry data have been deposited to the jPOST repository[66] under the accession numbers JPST001505 (anti-FLAG IP) and JPST001502 (anti-GFP IP). These data are also available via ProteomXchange under accession codes PXD031938 and PXD031917. The processed mass spectrometry data are provided in Supplementary Data 1 and 2. Atomic coordinates used in this study are available in the Protein Data Bank under accession codes 2RT5, 1OW1, 6QV2, 6IC8, and 5KXF and in the Alpha Fold Protein Structure Database under accession codes Q9BTC0, Q6ZMY3, and Q96T37. Source data are provided in this paper.

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

## Acknowledgements

We thank Anton Meinhart and Renato Arnese for RBM15 SPOC X-ray data collection; Martin Puchinger for help with labeling CTD peptides and advice regarding fluorescence anisotropy data analysis; Felix Fischer, Kathrin Nagl, and Alexander Athanasiadis for help with purifying SPOC domains and FA assays; Dirk Eick for sharing ZNF768 antibody; the VBCF Next Generation Sequencing facility for sequencing; the Max Perutz Labs Mass Spectrometry Facility, Markus Hartl, Dorothea Anrather and Natascha Hartl for mass spectrometry sample processing, data acquisition, and analysis; the staff of the X-ray beamlines at ESRF Grenoble and DSL Didcot for their support. L.A. was a member of the Integrative Structural Biology Ph.D. program funded by the Austrian Science Fund (W1258 "DK: Integrative Structural Biology") from 2018-2021. K.D.C. research was supported by the Wellcome Trust Collaborative Award (201543/Z/16); COST action BM1405—Non-globular proteins—from sequence to structure, function and application in molecular physio-pathology (NGP-NET); WWTF (Vienna Science and Technology Fund) Chemical Biology project LS17-008; Christian Doppler Laboratory for High-Content Structural Biology and Biotechnology; the Austrian-Slovak Interreg Project B301 StruBioMol, University of Vienna Research Platform Comammox and by the University of Vienna. B.Z. is supported by the Austrian Science Fund (P30550). This work was funded by the Austrian Science Fund (P31546 and W1258 "DK: Integrative Structural Biology" to D.S.).

## Author contributions

L.A. purified SPOC domains, labeled CTD peptides, performed and analyzed fluorescence anisotropy assays, generated endogenously tagged SHARP and PHF3 cell lines, performed RNA-seq, TT$_{chem}$-seq, and ChIP-seq experiments, performed and analyzed ChIP-qPCR experiments, performed SEC-MALLS experiments and wrote the manuscript. V.F. designed and performed the analysis of NGS data. J.B. generated DIDO cell lines, performed and analyzed co-immunoprecipitation assays, analyzed mass spectrometry data, and performed RNA-seq experiments. I.G. solved the SPOC structures. X.S. performed RNA-seq experiments. A.P. performed HADDOCK modeling experiments. G.A. performed mass spectrometry analysis of nucleoside modifications. S.P. generated RBM15 cell lines. A.N. and A.W. purified SPOC domains and performed fluorescence anisotropy assays. L.W. performed immuno-fluorescence analysis. S.K. supervised mass spectrometry analysis of nucleoside modifications. B.Z. supervised modeling experiments. K.D.C. supervised X-ray analysis. A.A. designed and supervised sequencing data analysis. D.S. conceived the study, performed, supervised, analyzed experiments, and wrote the paper.

## Competing interests

The authors declare no competing interests.

## Additional information

[1]Department of Radiation Oncology, Medical University of Vienna, Währinger Gürtel 18-20, 1090 Vienna, Austria. [2]Comprehensive Cancer Center, Medical University of Vienna, Spitalgasse 23, 1090 Vienna, Austria. [3]Department of Medical Biochemistry, Medical University of Vienna, Max Perutz Labs, Vienna Biocenter, Dr. Bohr-Gasse 9, 1030 Vienna, Austria. [4]The Berlin Institute for Medical Systems Biology, Max Delbrück Center, Robert-Rössle-Straße 10, 13125 Berlin, Germany. [5]Vienna Biocenter PhD Program, a Doctoral School of the University of Vienna and Medical University of Vienna, 1030 Vienna, Austria. [6]Department of Structural and Computational Biology, Max Perutz Labs, University of Vienna, Vienna Biocenter, Campus Vienna Biocenter 5, 1030 Vienna, Austria. [7]Department of Pharmaceutical Chemistry, Goethe University Frankfurt, Max-von-Laue-Straße 9, 60438 Frankfurt, Germany. [8]Department of Biochemistry, Faculty of Chemistry and Chemical Technology, University of Ljubljana, Večna Pot 113, 1000 Ljubljana, Slovenia. [9]European Molecular Biology Laboratory (EMBL) Grenoble, 71 Avenue des Martyrs, CS 90181, 38042 Grenoble, Cedex 9, France. ✉e-mail: dea.slade@maxperutzlabs.ac.at

