## [Peer Review File · Nature Communications]

The SPOC domain is a phosphoserine binding module that bridges transcription machinery with co- and post-transcriptional regulatorsREVIEWER COMMENTS

Reviewer #1 (Remarks to the Author):

Review Appel et al,

In this article, the authors have investigated the structure and function of various SPOC domains found in human proteins, including PHF3, DIDO, SHARP and RBM15. The PHF3 and DIDO share common specific structural features (PHD and TLD domains) while SHARP and RBM15 have in common the presence of RNA binding motifs. In the first part of the manuscript Appel et al, describe that while PHF3 and DIDO can bind the CTD domain of RNA Polymerase II by docking on the P-Ser2 peptides, both SHARP and RBM15 have affinity for P-Ser5, suggesting possible roles relating to the control of elongation and initiation of transcription, respectively. In the second part of the manuscript, the authors perform mass spectrometry analyses by overexpressing the 4 individual SPOC domains allowing to isolate new potential interacting proteins of these domains. Some of these interactions are confirmed in Co-IP experiments with the full-length proteins in the presence or absence of the SPOC domains. From these, it appears that PHF3 and DIDO tend to associate with regulators of elongation in a SPOC dependent manner. In the case of SHARP and RBM15, interactions with m6A writers and readers are lost when SPOC is absent. Mass spectrometry were also performed in the context of SHARP or SHARP- Δ SPOC expressed from endogenous locus, showing a loss of interaction to the FMR m6A binder. Finally, genome-wide location analyses of SHARP and PHF3 were performed and are presented in Fig. 6. This study echoes a recent article from in which the authors showed the important role of PHF3 and its SPOC in neurogenesis as a major Pol II CTD regulator.

Overall, the experiments are well performed, especially the structural and biochemical parts of the manuscript and the insights into the SPOC domain functions are interesting for a specialized audience. Many novel interactors are described and it is suggested that CTD and Pol II elongation might represent common targets to these domains. However, the manuscript remains overall descriptive and little information as to the function of the SPOC domains in DIDO, SHARP and RBM15 for transcription and/or RNA processing is provided. Furthermore, the genome-wide analyses of SHARP and PHF3 are superficial and do not yield clear conclusions.

Major remarks

- 1- One major conclusion of this article is that all 4 SPOC domains display the ability to bind phosphorylated CTD peptides through conserved basic patches, similar to the one already described by this group in PHF3. The proteins bearing these domains also interact with Pol II and in some cases with elongation proteins. No further experiment is provided to indicate whether the perturbation of these proteins (for example through knock-down) would somehow perturb transcriptional activity in initiation or elongation. The nascent transcriptome or Pol II ChIP (using P-CTD or total Pol II) could help confirming that DIDO/PHF3 relate to elongation control while RBM15/SHARP relate to initiation.
- 2- With the same concern in mind, and since the authors have built SPOC mutants for SHARP and PHF3 (GFP fusions). Do these show any cellular or molecular phenotypes relating to transcription? While the authors already studied PHF3 in the past, further insights in this cellular context and compared to that of SHARP would improve the general interest of the study.
- 3- The association of SHARP/RBM15 to m6A processing is also not functionally demonstrated. Does SHARP impairment impact on m6A or RNA stability (nascent transcript vs stable transcripts)?
- 4- The ChIP-seq experiment depicted in Fig. 6 is not convincing and the data shown is not bringing any possible conclusion. How many peaks are isolated in the WT and mutant backgrounds? Where are they located? In the case of PHF3, how do they relate to the recently published study of the authors? How can SHARP ChIP-seq yield a unique ChIP signal over the whole genome? To the least a differential analysis (DEseq or other) should be performed between the full-length and the SPOC-truncated versions of SHARP and PHF3. Based on the method description it is not possible to determine how many replicates were performed, what are the control quality applied? It is perfectly understandable that a ChIP experiment may not yield interpretable signal for multiple reasons. In this

case and relating to the point 1- above, a Pol II ChIP experiment in mutant or kd background could be an alternative.

Minor remarks

1- Strength of wording.

a. In the abstract, last sentence: Our findings establish.... 'suggest' would more appropriate

b. Discussion, first paragraph: This indicates that SHARP.... 'suggests' would more appropriate

2- Introduction, 3rd paragraph: The disordered CTD comprises up to 52 heptarepeats of the sequence YSPTSPS and is differentially modified throughout the transcription cycle. The term 'imperfect' may be added before heptarepeats.

3- Result section p6: However, SPOC deletion did not impair...(Fig. 5e). I guess that this should indicate SPOC deletion 'in SHARP'.

Reviewer #2 (Remarks to the Author):

This manuscript from the group of Dea Slade and co-workers describes how SPOC domains from different proteins interact with phosphorylated serine residues of different substrate binders. They use comparative crystal structure analyses, fluorescence anisotropy binding measurements, and proteomic interactome studies to describe the specificities of the various mammalian SPOC domains. The authors determine the genomic localization of SPOC domains to conclude with a model, how these domains may combine transcription with co- and post-translational processes including N6-methyladenosine RNA modifications.

In this study, the authors establish SPOC domains as a RNA pol II CTD reader domain that recognizes different CTD phosphorylation patterns in tandem hepta-repeats. However, the manuscript does not appear well structured and some of the results are difficult to grasp. I have a couple of critical points that I would like to see addressed before considering the manuscript as strong candidate for publication.

Criticism:

The abstract can be improved, e.g., m6A is not introduced. "We report the crystal structure of SHARP (SPEN) SPOC-CTD and identify the molecular determinants for its specific binding to phosphorylated serine-5." What is SPEN? It is not intuitively clear that SPOC-CTD refers to a protein-ligand peptide complex structure. The third sentence might be shortened or included into the following, while the number of protein names and abbreviations in the remainder is confusing. Please rewrite.

The description of basic surface patches in Figure 1 is misleading. I understand that all SPOC domains are shown in the same orientation, but why is "Patch 2" in PHF3 switched to "Patch 1" in SHARP and RBM15 (compare panel b with panels c and d, bottom right)? I suggest to choose two moderately different colors, e.g., blue and light blue, for the alignment of Fig. 1a and label the basic residues that constitute these patches in columns. This would be easy to understand and one can immediately identify the amino acid differences between the different SPOC domains. As SHARP SPOC has only one basic patch, why not naming this "patch 1" and the second optional patch in then "patch 2"? Or leave patch 1 out for SHARP.

The secondary structure elements should be shown above the sequence alignment in Fig. 1a and residue numbers provided such that one can at least for one of the following four structures (panels b to e) correlate the residues with the labelled surface patches.

When providing the distances between the basic surface patches on page 3 and in Fig. legend 1,

specifying three decimal places seems inappropriate to me. Whole numbers would be quite sufficient.

The first data paragraph in the Results section is very long. I suggest to split it into two following also the description of Figures 1 and 2.

The manuscript would benefit from a discussion of the oligomerization properties of SPOC domain-containing proteins. There could be a potential similarity to bromodomain-containing proteins that bind as reader domains to acetylated-Lys residues of the histone code with low affinities. However, in Brd-containing proteins oligomerization and the assembly of multiple bromodomains lead to cooperative effects that ultimately render the recognized sequences specific. A display of the molecular domain architecture and the possible oligomerization state of the SPOC domain-containing proteins could be thus informative to the reader.

Fig. 3: Did the authors also test a doubly pS5 phosphorylated CTD tandem repeat for crystallization, similarly designed as the peptide used in the PHF3-2xS2P crystal structure? This would make it a TpS5PSYSPTpS2PS peptide?

Along this line: The authors should discuss the different binding modalities of the tyrosine residues in the PHF3 SPOC domain (Fig. 1b) with the SHARP SPOC domain (Fig. 3d). I would assume that this determines the register of the tandemly phosphorylated CTD peptide recognition.

We thank the reviewers for their critical feedback and provide our point-by-point response below. The revised manuscript contains many additional experiments that bring new important insights into the interactions and functions of SPOC domain proteins. All the changes in the manuscript text and figures are highlighted in yellow.

REVIEWER COMMENTS

Reviewer #1 (Remarks to the Author):

Review Appel et al,

In this article, the authors have investigated the structure and function of various SPOC domains found in human proteins, including PHF3, DIDO, SHARP and RBM15. The PHF3 and DIDO share common specific structural features (PHD and TLD domains) while SHARP and RBM15 have in common the presence of RNA binding motifs. In the first part of the manuscript Appel et al, describe that while PHF3 and DIDO can bind the CTD domain of RNA Polymerase II by docking on the P-Ser2 peptides, both SHARP and RBM15 have affinity for P-Ser5, suggesting possible roles relating to the control of elongation and initiation of transcription, respectively. In the second part of the manuscript, the authors perform mass spectrometry analyses by overexpressing the 4 individual SPOC domains allowing to isolate new potential interacting proteins of these domains. Some of these interactions are confirmed in Co-IP experiments with the full-length proteins in the presence or absence of the SPOC domains. From these, it appears that PHF3 and DIDO tend to associate with regulators of elongation in a SPOC dependent manner. In the case of SHARP and RBM15, interactions with m6A writers and readers are lost when SPOC is absent. Mass spectrometry were also performed in the context of SHARP or SHARP-DSPOC expressed from endogenous locus, showing a loss of interaction to the FMR m6A binder. Finally, genome-wide location analyses of SHARP and PHF3 were performed and are presented in Fig. 6. This study echoes a recent article from in which the authors showed the important role of PHF3 and its SPOC in neurogenesis as a major Pol II CTD regulator.

Overall, the experiments are well performed, especially the structural and biochemical parts of the manuscript and the insights into the SPOC domain functions are interesting for a specialized audience. Many novel interactors are described and it is suggested that CTD and Pol II elongation might represent common targets to these domains. However, the manuscript remains overall descriptive and little information as to the function of the SPOC domains in DIDO, SHARP and RBM15 for transcription and/or RNA processing is provided. Furthermore, the genome-wide analyses of SHARP and PHF3 are superficial and do not yield clear conclusions.

Major remarks

1- One major conclusion of this article is that all 4 SPOC domains display the ability to bind phosphorylated CTD peptides through conserved basic patches, similar to the one already described by this group in PHF3. The proteins bearing these domains also interact with Pol II and in some cases with elongation proteins. No further experiment is provided to indicate whether the perturbation of these proteins (for example through knock-down) would somehow perturb transcriptional activity in initiation or elongation. The nascent transcriptome or Pol II ChIP (using P-CTD or total Pol II) could help confirming that DIDO/PHF3 relate to elongation control while RBM15/SHARP relate to initiation.

To address this, we analysed the nascent transcriptome of SPOC protein knock-out cell lines and endogenous Δ SPOC mutants by transient transcriptome sequencing (TT_{chem}-seq). We chose TT_{chem}-seq over Pol II ChIP-seq because it provides (i) higher resolution than ChIP and (ii) information on actual transcriptional output rather than just the position of the polymerase.

We generated DIDO KO and DIDO Δ SPOC as well as RBM15 KO and RBM15 Δ SPOC HEK293T cell lines. We had previously generated PHF3 KO and PHF3 Δ SPOC (Appel et al., 2021). Despite our best efforts we were unable to generate a SHARP KO cell line, suggesting that SHARP deletion is lethal in HEK293 cells. Information on new cell line generation and validation is included in Supplementary Figs. 10 and 11.

TT_{chem}-seq was performed on the following samples:

- PHF3 WT, KO and Δ SPOC
- DIDO WT, KO and Δ SPOC
- SHARP-GFP and SHARP Δ SPOC-GFP
- RBM15 WT, KO and Δ SPOC

We confirmed our previous results (Appel et al., 2021) that nascent transcripts are globally reduced in PHF3 KO and Δ SPOC and that the stalling index (TSS/gene body reads) is slightly increased, indicating that PHF3 positively regulates transcription elongation. DIDO Δ SPOC showed a major upregulation of nascent transcripts and a slight reduction in stalling index, suggesting that DIDO negatively regulates transcription elongation through its SPOC domain. RBM15 Δ SPOC showed a subtle reduction in nascent transcripts and a subtle increase in stalling, suggesting that RBM15 acts as a weak positive regulator of transcription elongation. In SHARP Δ SPOC nascent transcripts were equally reduced at TSS and gene bodies, without a change in stalling index, suggesting that SHARP positively regulates transcription initiation. Thus we can conclude that PHF3 and DIDO regulate transcription elongation through SPOC-mediated recognition of pSer2 CTD, SHARP regulates transcription initiation through SPOC-mediated recognition of pSer5 CTD, whereas RBM15 weakly regulates transcription and acts predominantly as a regulator of mRNA stability (see responses to points 2 and 3). These results are included in new Fig. 7 and Supplementary Dataset 4.

Fig. 7: SPOC domain proteins regulate transcription and mRNA stability. **a** Genome-wide distribution of log₂ fold changes of TT_{chem}-seq signal in KO and ΔSPOC relative to WT on **a** TSS and **b** gene body regions. Cells were treated with 1 mM 4sU for 15 min. In vitro transcribed synthetic 4sU-labeled RNA and 4sU-labeled yeast RNA were used for spike-in normalization. Experiments were performed in three independent replicates. **c** Stalling index analysis calculated as TT_{chem}-seq TSS/gene body signal. **d** Density distribution of the differences in log₂ fold changes KO/WT or ΔSPOC/WT between RNA-seq and TT_{chem}-seq data.

2- With the same concern in mind, and since the authors have built SPOC mutants for SHARP and PHF3 (GFP fusions). Do these show any cellular or molecular phenotypes relating to transcription? While the authors already studied PHF3 in the past, further insights in this cellular context and compared to that of SHARP would improve the general interest of the study.

In addition to analysing nascent transcripts by TT_{chem}-seq, we analysed mature transcripts by RNA-seq. We previously published RNA-seq in PHF3 KO and PHF3 ΔSPOC (Appel et al., 2021). Additionally, we performed RNA-seq in:

- DIDO WT, KO and ΔSPOC
- SHARP-GFP and SHARP ΔSPOC-GFP
- RBM15 WT, KO and ΔSPOC

We previously showed that mature transcripts are upregulated in PHF3 KO and Δ SPOC (Appel et al., 2021). In contrast, transcripts were predominantly downregulated in DIDO and RBM15 KO and Δ SPOC, with RBM15 KO/ Δ SPOC having a stronger effect (Fig. 6). In SHARP Δ SPOC only a small number of transcripts were downregulated. The integration of RNA-seq and TT_{chem}-seq data revealed that PHF3, DIDO and RBM15 regulate mRNA stability, with PHF3 acting as a negative regulator and DIDO/RBM15 as positive regulators of mRNA stability (Fig. 7d).

These results are included in new Fig. 6 and Supplementary Dataset 3.

Fig. 6: SPOC domain proteins regulate gene expression and m6A levels. **a, b, d, e, g, h, i** MA plots showing RNA-seq log₂ fold change (KO/WT or Δ SPOC/WT) versus log₁₀ mean expression in WT for **a** PKF3 KO, **b** PHF3 Δ SPOC, **d** DIDO KO, **e** DIDO Δ SPOC, **g** RBM15 KO, **h** RBM15 Δ SPOC and **i** SHARP Δ SPOC. Red and blue dots indicate upregulated and downregulated genes respectively with fold-change>2, p<0.05. *Drosophila* S2 cells were used for spike-in normalization. **c, f, j** Venn diagram showing overlaps between **c** upregulated genes in PHF3 KO and PHF3 Δ SPOC, **f** downregulated genes in

DIDO KO and DIDO Δ SPOC, **j** downregulated genes in SHARP Δ SPOC, RBM15 KO and RBM15 Δ SPOC. **k** m6A levels are decreased upon impairment of RBM15. Mass spectrometry analysis of single nucleosides derived from mRNA isolated from the indicated cell lines. Data are presented as mean \pm standard deviation of three replicates, individual data points are indicated as black dots. One-tailed, two-sample equal variance t-test was used to determine p-values.

3- The association of SHARP/RBM15 to m6A processing is also not functionally demonstrated. Does SHARP impairment impact on m6A or RNA stability (nascent transcript vs stable transcripts)?

In collaboration with Stefanie Kaiser (Goethe University Frankfurt) we analysed m6A levels in mRNA isolated from RBM15 WT, RBM15 KO, RBM15 Δ SPOC, SHARP-GFP WT and SHARP Δ SPOC-GFP HEK293T cells by mass spectrometry as described in Borland et al. 2018 (DOI: 10.3390/genes10010026).

The analysis revealed a prominent decrease of m6A modification in mRNA from RBM15 KO and RBM15 Δ SPOC with no change upon SPOC deletion in SHARP (Fig. 6k). This indicates that RBM15 and its SPOC domain are critical for m6A modification of mRNA, while SHARP is dispensable, which is not surprising given that it interacts with the m6A reader FMR1 rather than the m6A writer complex and is thus more likely to influence the fate of m6A-modified RNAs rather than establishment of the modification itself.

Interestingly, the same cell lines that show reduced m6A levels also show a striking downregulation of mature transcripts in RNA-seq (Fig. 6g,h), which suggests a stabilizing effect of m6A modification deposited through RBM15.

To additionally characterize the interaction between the RBM15 SPOC domain and the m6A writer complex, we analysed the binding of RBM15 SPOC to (phospho-)peptides derived from the m6A writer complex component WTAP, which was identified as an interactor of the RBM15 SPOC domain by mass spectrometry and a SPOC-dependent interactor of full-length RBM15 by Co-IP and western blotting. We determined the binding affinity of RBM15 SPOC to a peptide corresponding to the first 19 N-terminal amino acids of WTAP (Fig. 5f). RBM15 SPOC bound to the peptide phosphorylated at S14 with an affinity of $21.5 \pm 1.5 \mu\text{M}$ but did not bind to the unphosphorylated peptide. Mutation of the conserved residue R834 to alanine reduced the binding affinity to $50.4 \pm 1.1 \mu\text{M}$, but did not abrogate the binding, indicating that R834 is a major contributor but not the only determining factor for recognition of WTAP pS14.

Fig. 5f: Fluorescence anisotropy binding curves of RBM15 SPOC with WTAP peptides. Fluorescence anisotropy is plotted as a function of protein concentration. Data points and error bars show mean anisotropy \pm standard deviation from 3 independent experiments.

Overall, these experiments show that RBM15 directly binds to phosphorylated WTAP in a SPOC-dependent manner to guide the m6A writer complex onto target RNAs. While RRM of RBM15 were known to bind to target RNA, it was not clear how RBM15 recruits the m6A writer complex. Our findings that RBM15 recruits the m6A writer complex in a SPOC-dependent manner provide an

important contribution to the m6A field. Moreover, given that RBM15 loss or SPOC deletion results in globally reduced cellular m6A levels and globally reduced transcripts levels, RBM15 may positively regulate mRNA stability through the m6A RNA modification. The m6A RNA modification was previously shown to regulate mRNA decay and translation and may exert a positive or a negative effect on mRNA stability depending on m6A reader proteins. We hypothesize that RBM15 can selectively guide the m6A writer complex to locations where the m6A mark promotes mRNA stability.

4- The ChIP-seq experiment depicted in Fig. 6 is not convincing and the data shown is not bringing any possible conclusion. How many peaks are isolated in the WT and mutant backgrounds? Where are they located? In the case of PHF3, how do they relate to the recently published study of the authors? How can SHARP ChIP-seq yield a unique ChIP signal over the whole genome? To the least a differential analysis (DEseq or other) should be performed between the full-length and the SPOC-truncated versions of SHARP and PHF3. Based on the method description it is not possible to determine how many replicates were performed, what are the control quality applied? It is perfectly understandable that a ChIP experiment may not yield interpretable signal for multiple reasons. In this case and relating to the point 1- above, a Pol II ChIP experiment in mutant or kd background could be an alternative.

Previously we performed one ChIP-seq replicate and validated a few loci by ChIP-qPCR. To allow for a more robust analysis of PHF3 and SHARP genomic occupancy, we now performed additional ChIP-seq replicates to have three replicates in total for PHF3 and two for SHARP. The signal distribution of the ChIP data is similar to Pol II profiles, with enrichment of large genomic regions. In such cases, peak calling requires extensive optimization. We have therefore performed gene-based quantification that showed reduced chromatin association for PHF3 Δ SPOC on genes that are deregulated in PHF3 Δ SPOC according to RNA-seq. For SHARP we could not detect major changes other than the Xist locus, which is not surprising considering that i) SHARP binding to Pol II is weaker compared to PHF3 and ii) SHARP is an RNA-binding protein. This is included in the manuscript text, Supplementary Data 5 and Supplementary Fig. 13.

Fig. S13: ChIP-seq analysis of the changes in chromatin occupancy upon loss of the SPOC domain in PHF3 and SHARP in HEK293T cells. a,c Relationship between ChIP-seq body fold change and ChIP-seq TSS fold change for a PHF3 Δ SPOC-

GFP vs PHF3-GFP WT (N=3) and c SHARP Δ SPOC-GFP vs SHARP-GFP WT (N=2). Blue and red dots indicate genes with reduced or increased genomic occupancy respectively. **b,d** Relationship between RNA-seq fold change and ChIP-seq body fold change for b PHF3 Δ SPOC vs PHF3 WT and d SHARP Δ SPOC vs SHARP WT. Blue and red dots indicate genes with reduced or increased RNA-seq gene expression levels respectively.

Minor remarks

1- Strength of wording.

- a. In the abstract, last sentence: Our findings establish.... 'suggest' would more appropriate
- b. Discussion, first paragraph: This indicates that SHARP.... 'suggests' would more appropriate

2- Introduction, 3rd paragraph: The disordered CTD comprises up to 52 heptarepeats of the sequence YSPTSPS and is differentially modified throughout the transcription cycle. The term 'imperfect' may be added before heptarepeats.

3- Result section p6: However, SPOC deletion did not impair...(Fig. 5e). I guess that this should indicate SPOC deletion 'in SHARP'.

We incorporated the suggested changes into the manuscript.

Reviewer #2 (Remarks to the Author):

This manuscript from the group of Dea Slade and co-workers describes how SPOC domains from different proteins interact with phosphorylated serine residues of different substrate binders. They use comparative crystal structure analyses, fluorescence anisotropy binding measurements, and proteomic interactome studies to describe the specificities of the various mammalian SPOC domains. The authors determine the genomic localization of SPOC domains to conclude with a model, how these domains may combine transcription with co- and post-translational processes including N6-methyladenosine RNA modifications.

In this study, the authors establish SPOC domains as a RNA pol II CTD reader domain that recognizes different CTD phosphorylation patterns in tandem hepta-repeats. However, the manuscript does not appear well structured and some of the results are difficult to grasp. I have a couple of critical points that I would like to see addressed before considering the manuscript as strong candidate for publication.

Criticism:

The abstract can be improved, e.g., m6A is not introduced. "We report the crystal structure of SHARP (SPEN) SPOC-CTD and identify the molecular determinants for its specific binding to phosphorylated serine-5." What is SPEN? It is not intuitively clear that SPOC-CTD refers to a protein-ligand peptide complex structure. The third sentence might be shortened or included into the following, while the number of protein names and abbreviations in the remainder is confusing. Please rewrite.

We adapted the abstract to include these suggestions.

The description of basic surface patches in Figure 1 is misleading. I understand that all SPOC domains are shown in the same orientation, but why is "Patch 2" in PHF3 switched to "Patch 1" in SHARP and RBM15 (compare panel b with panels c and d, bottom right)? I suggest to choose two moderately different colors, e.g., blue and light blue, for the alignment of Fig. 1a and label the basic residues that

constitute these patches in columns. This would be easy to understand and one can immediately identify the amino acid differences between the different SPOC domains. As SHARP SPOC has only one basic patch, why not naming this “patch 1” and the second optional patch in then “patch 2”? Or leave patch 1 out for SHARP.

We marked the conserved residues in the alignment in Fig. 1a in different colours for the individual patches and marked the corresponding patches in Fig. 1b-f in the same colours. We hope that this provides more clarity and makes it easy to understand which residues correspond to which patch.

We also renamed the patches in Fig. 1b, d and f, such that “Patch 1” is consistent between all SPOC domains.

The secondary structure elements should be shown above the sequence alignment in Fig. 1a and residue numbers provided such that one can at least for one of the following four structures (panels b to e) correlate the residues with the labelled surface patches.

We added the secondary structure elements above the primary sequence in the alignment. Residue numbers are provided at the end of each line. We additionally marked the highly conserved Arg residue (conserved in all SPOC domains but SPOCD1) with a red asterisk in the labels of Fig 1b-f to make it easier to correlate the residues with the patches.

When providing the distances between the basic surface patches on page 3 and in Fig. legend 1, specifying three decimal places seems inappropriate to me. Whole numbers would be quite sufficient.

We rounded the distances to whole numbers in the figure legend and the main text.

The first data paragraph in the Results section is very long. I suggest to spilt it into two following also the description of Figures 1 and 2.

We split the first paragraph into two sections: *Conserved basic residues cluster to patches on the surface of SPOC* and *Basic surface patches mediate SPOC binding to phosphorylated serine*.

The manuscript would benefit from a discussion of the oligomerization properties of SPOC domain-containing proteins. There could be a potential similarity to bromodomain-containing proteins that bind as reader domains to acetylated-Lys residues of the histone code with low affinities. However, in Brd-containing proteins oligomerization and the assembly of multiple bromodomains lead to cooperative effects that ultimately render the recognized sequences specific. A display of the molecular domain architecture and the possible oligomerization state of the SPOC domain-containing proteins could be thus informative to the reader.

We added an overview of the domain architecture of SPOC proteins to the manuscript (New supplementary Fig. 1). To determine whether SPOC domains are monomeric or form oligomers, we performed SEC-MALS of the SPOC domains of PHF3, DIDO, SHARP and RBM15 (Supplementary Fig. 2). DIDO and SHARP SPOC were exclusively monomeric, while RBM15 showed a small percentage of dimer. PHF3 SPOC was present as a monomer (70%), but also formed dimers and trimers (30%). However, full-length PHF3 is monomeric (Appel et al, 2021).

Fig. 3: Did the authors also test a doubly pS5 phosphorylated CTD tandem repeat for crystallization, similarly designed as the peptide used in the PHF3-2xS2P crystal structure? This would make it a TpS5PSYSPTpS2PS peptide?

We did not test the 2xS5P peptide for crystallization. Since the affinity for the 1xS5P and 2xS5P peptides is essentially the same (Fig. 2) we did not expect a difference in the mode of binding or additional contacts between SHARP SPOC and the second phosphorylated Ser5 residue.

Along this line: The authors should discuss the different binding modalities of the tyrosine residues in the PHF3 SPOC domain (Fig. 1b) with the SHARP SPOC domain (Fig. 3d). I would assume that this determines the register of the tandemly phosphorylated CTD peptide recognition.

We added the following paragraph to the results section:

Tyrosine residues play an important role in phosphoserine recognition and determining the register of CTD-phosphoserine recognition. The conserved residue Y1291 in PHF3 SPOC/Y3602 in SHARP SPOC is involved in coordinating pS2 of the second CTD repeat in the case of PHF3 or pS5 in the case of SHARP SPOC. In addition, PHF3 residues Y1257 and Y1312 together with T1253 form a hydrophobic pocket for P6 of the first CTD repeat (Fig. 3f). These hydrophobic contacts mediate tight packing of the CTD peptide to the PHF3 SPOC surface and specific recognition of tandem S2 phosphorylation. Y1257 and Y1312 are not conserved in SHARP, which may explain the different mode of binding and why SHARP SPOC recognizes a single rather than two adjacent phosphorylated CTD repeats.

REVIEWERS' COMMENTS

Reviewer #1 (Remarks to the Author):

The authors have answered to most of my initial concerns in review. In my opinion the manuscript can now be published in its revised version.

Reviewer #2 (Remarks to the Author):

This is a very elaborate revision that addresses all my previous concerns. Although not raised by this reviewer, I particularly appreciate the new TT-seq data with the four SPOC domains PHF3, DIDO, SHARP, and RBM15 illustrating their impact in transcription regulation and mRNA stability. I have no further comments and suggestions to the present form of the manuscript.